# Fed-SB: A *Silver Bullet* for Extreme Communication Efficiency and Performance in (Private) Federated LoRA Fine-Tuning

**Raghav Singhal**[*1], **Kaustubh Ponkshe**[*1], **Rohit Vartak**[2], **Lav R. Varshney**[3], and **Praneeth Vepakomma**[1,4]

[1]Mohamed bin Zayed University of Artificial Intelligence, UAE
[2]Duke University, USA
[3]University of Illinois Urbana-Champaign, USA
[4]Massachusetts Institute of Technology, USA

**Reviewed on OpenReview:** https://openreview.net/forum?id=87UyFEhzyP

## Abstract

Low-Rank Adaptation (LoRA) has become ubiquitous for efficiently fine-tuning foundation models. However, federated fine-tuning using LoRA is challenging due to suboptimal updates arising from traditional federated averaging of individual adapters. Existing solutions either incur prohibitively high communication cost that scales linearly with the number of clients or suffer from performance degradation due to limited expressivity. We introduce **Federated Silver Bullet (Fed-SB)**, a novel approach for federated fine-tuning of LLMs using LoRA-SB, a recently proposed low-rank adaptation method. LoRA-SB optimally aligns the optimization trajectory with the ideal low-rank full fine-tuning projection by learning a small square matrix ($R$) between adapters $B$ and $A$, keeping other components fixed. Direct averaging of $R$ guarantees exact updates, substantially reducing communication cost, which remains independent of the number of clients, and enables scalability. Fed-SB achieves state-of-the-art performance across commonsense reasoning, arithmetic reasoning, and language inference tasks while reducing communication costs by up to 230x. In private settings, Fed-SB further improves performance by (1) reducing trainable parameters, thereby lowering the noise required for differential privacy and (2) avoiding noise amplification introduced by other methods. Overall, Fed-SB offers a state-of-the-art, efficient, and scalable solution for both private and non-private federated fine-tuning. Our code is publicly available at: https://github.com/CERT-Lab/fed-sb.

## 1 Introduction

Large language models (LLMs) have demonstrated remarkable generalization across a wide range of tasks (2; 49; 46; 40). Fine-tuning (FT) remains the most effective approach for aligning LLMs to specific data distributions and reinforcing desired properties. However, as model sizes scale, full FT becomes increasingly prohibitive due to its substantial computational cost. To address this, parameter-efficient fine-tuning (PEFT) techniques, such as low-rank adaptation (LoRA, (21)), have emerged as viable alternatives, offering a favorable trade-off between computational efficiency and performance. Variants of LoRA, including QLoRA (14), DoRA (32), AdaLoRA (60), and LoRA-SB (39), further refine this paradigm by optimizing memory efficiency, training dynamics, and generalization.

Federated learning (FL) is a popular method for training models in settings where data is siloed across multiple entities (26; 24; 7). Federated FT extends this paradigm by enabling large models, pre-trained on

---

[*]Equal Contributions. Author ordering determined randomly.

public data, to be efficiently adapted to private, distributed datasets without requiring clients to share their local data. Existing methods predominantly rely on LoRA-based techniques to learn client-specific adaptations (58). However, optimizing federated aggregation often involves tradeoffs between model performance (44) and communication efficiency (52; 43), necessitating careful design choices to balance these competing objectives.

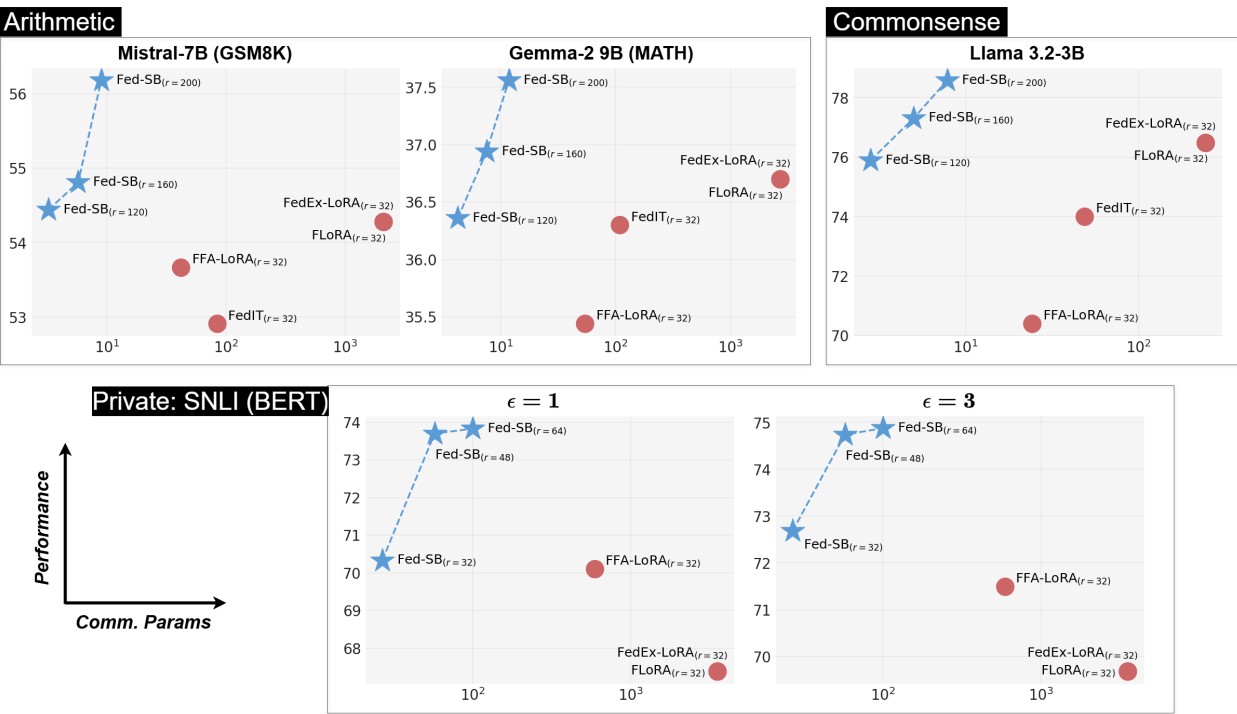

Figure 1: Performance vs. communicated parameter cost (log scale) for Fed-SB and other federated fine-tuning methods in both non-private and privacy-preserving federated settings. Fed-SB achieves state-of-the-art accuracy while significantly reducing communication cost. Communicated parameters are in thousands for BERT and millions for other models.

LoRA-SB (39), optimally simulates full fine-tuning in low-rank spaces by learning an $r \times r$ matrix between the low-rank adapters **A** and **B** while keeping other components fixed. This design reduces trainable parameters and enables better updates through its initialization strategy. Moreover, LoRA-SB demonstrates that this optimal approximation is not achievable with standard LoRA-based methods. LoRA-SB learns higher-rank updates with 2–4x greater rank than LoRA while requiring 45-90x fewer parameters. We propose **Fed-SB**, a federated variant of LoRA-SB, providing an ideal framework for (private) federated FT. Fed-SB overcomes limitations in LoRA-based federated FT while being significantly more computation- and communication-efficient. Notably, it enables exact and optimal aggregation by simply averaging the learnable matrix **R**.

Differential privacy (DP) is a well-established framework for ensuring strong privacy guarantees (17; 18), which is particularly crucial in federated settings. DP-SGD is a widely used privacy-preserving optimization method (1), but its challenges are exacerbated in federated FT, where noise injected for privacy amplifies divergence across client models (44). Learning in DP-SGD is more effective when the number of learnable parameters is reduced, as the magnitude of noise added for privacy guarantees scales with the parameter count. Fed-SB mitigates this issue to yield improved performance, since it inherently has fewer learnable parameters and thus less noise injection. Furthermore, we show that Fed-SB avoids noise amplification introduced by other methods, further enhancing privacy-preserving learning.

Fed-SB offers an extremely efficient and scalable solution for both private and non-private federated FT, as shown in Figure 1. It consistently has superior performance while substantially reducing communication overhead than other methods. Our key contributions are summarized as follows:

- We propose **Fed-SB**, a federated fine-tuning method that achieves exact and optimal aggregation in low-rank adaptation without incurring prohibitive communication costs or performance degradation.
- Fed-SB consistently achieves state-of-the-art results while significantly reducing communication cost, by up to 230x, by requiring only an $r \times r$ matrix to be transmitted per aggregation.
- We demonstrate that Fed-SB is particularly well-suited for privacy-preserving (federated) fine-tuning, as it minimizes noise by reducing the number of learnable parameters and leveraging linearity in the aggregate update.
- Extensive experiments on 4 models across 3 diverse benchmarks show that Fed-SB consistently outperforms existing methods while drastically reducing communication overhead in both private and non-private federated settings.

Table 1: Advantages of Fed-SB over various SOTA federated fine-tuning methods ($c$ clients). Fed-SB achieves exact aggregation and high expressivity with extremely low communication cost - constant with the number of clients. In private settings, Fed-SB offers additional advantages by minimizing noise through reducing learnable parameters and leveraging linearity to avoid noise amplification.

| | FedIT | FLoRA | FedEx-LoRA | FFA-LoRA | Fed-SB |
|---|---|---|---|---|---|
| Exact aggregation | ✗ | ✓ | ✓ | ✓ | ✓ |
| Learnable params. | $\mathcal{O}((m+n)r)$ | $\mathcal{O}((m+n)r)$ | $\mathcal{O}((m+n)r)$ | $\mathcal{O}(mr)$ | $\mathcal{O}(r^2)$ |
| Communication cost | $\mathcal{O}((m+n)r)$ | $\mathcal{O}(min(c(m+n)r, mn))$ | $\mathcal{O}(min(c(m+n)r, mn))$ | $\mathcal{O}(mr)$ | $\mathcal{O}(r^2)$ |
| No noise ampl. | ✗ | ✗ | ✗ | ✓ | ✓ |
| Privacy (less params.) | ✗ | ✗ | ✗ | ✗ | ✓ |
| Optimal expressivity | ✓ | ✓ | ✓ | ✗ | ✓ |

## 2 Preliminaries and Motivation

**Federated Fine-Tuning.** Given a pretrained weight matrix $\mathbf{W} \in \mathbb{R}^{m \times n}$, the objective in FT is to learn an update $\Delta\mathbf{W}$ for a given dataset. LoRA (21) remains the preferred method, where low-rank adapter matrices $\mathbf{A} \in \mathbb{R}^{r \times n}$ and $\mathbf{B} \in \mathbb{R}^{m \times r}$ are learned such that $\Delta\mathbf{W} = \mathbf{BA}$. In federated learning, the dataset is distributed across $c$ clients, and the goal is to learn $\Delta\mathbf{W}$ without sharing local data with a central server. To achieve this, each client learns its own adapter matrices $\mathbf{A}_i$ and $\mathbf{B}_i$. The server aggregates these updates to refine $\mathbf{W}$, along with globally beneficial representations of $\mathbf{A}$ and $\mathbf{B}$, ultimately producing a shared aggregate model $\mathbf{W}^{\mathrm{agg}}$. Next, each client continues the local FT process, followed by aggregation at the end of each round. This cycle repeats over multiple rounds. We summarize some of the state-of-the-art federated FT methods below.

**Fed-IT** (58) updates the adapters $\mathbf{A}$ and $\mathbf{B}$ using the standard FedAvg (35) algorithm:

$$\mathbf{A}^{\mathrm{agg}} = \frac{1}{c}\sum_{i=1}^{c}\mathbf{A}_i, \quad \mathbf{B}^{\mathrm{agg}} = \frac{1}{c}\sum_{i=1}^{c}\mathbf{B}_i. \tag{1}$$

**FedEx-LoRA** (43) follows the same aggregation but introduces an additional error correction matrix $\mathbf{W}_{\mathrm{err}}$ of rank $\min(cr, m, n)$:

$$\mathbf{W}_{\mathrm{err}} = \left(\frac{1}{c}\sum_{i=1}^{c}\mathbf{A}_i\mathbf{B}_i\right) - \left(\frac{1}{c}\sum_{i=1}^{c}\mathbf{A}_i\right)\left(\frac{1}{c}\sum_{i=1}^{c}\mathbf{B}_i\right). \tag{2}$$

**FLoRA** (52) follows the same principle as FedEx-LoRA but achieves it by stacking the adapter matrices, and reinitializes them randomly at the end of each communication round. **FFA-LoRA** (44) keeps $\mathbf{A}$ fixed while training (and aggregating) only $\mathbf{B}$ matrices.

$$\mathbf{B}^{\mathrm{agg}} = \frac{1}{c}\sum_{i=1}^{c}\mathbf{B}_i. \tag{3}$$

$$\underbrace{\tilde{\mathbf{W}}^{global} = \mathbf{W}_0 + \frac{1}{k}\sum_{i=1}^{k}\mathbf{B}_i \times \frac{1}{k}\sum_{i=1}^{k}\mathbf{A}_i}_{\text{Parameters after aggregation with LoRA + FedAvg (FedIT)}} \neq \underbrace{\mathbf{W}_0 + \frac{1}{k}\sum_{i=1}^{k}(\mathbf{B}_i\mathbf{A}_i) = \mathbf{W}^{global}}_{\text{Ideal parameters following model-averaging}} \tag{4}$$

**(Approximate) Differential Privacy.** DP, introduced by (17), is a widely adopted mathematical framework for privacy preservation. A randomized mechanism $\mathcal{M} : \mathcal{D} \to \mathcal{R}$, mapping a domain $\mathcal{D}$ to a range $\mathcal{R}$, satisfies $(\epsilon, \delta)$-differential privacy if, for any two adjacent inputs $d, d' \in \mathcal{D}$ and any subset of outputs $S \subseteq \mathcal{R}$, the following holds:

$$\Pr[\mathcal{M}(d) \in S] \leq e^\epsilon \Pr[\mathcal{M}(d') \in S] + \delta. \tag{5}$$

$$\mathbf{B}_i^{j+1} \leftarrow \frac{1}{k}\sum_{i=1}^{k}\mathbf{B}_i^j, \mathbf{A}_i^{j+1} \leftarrow \frac{1}{k}\sum_{i=1}^{k}\mathbf{A}_i^j, \mathbf{W_0}^{j+1} \leftarrow \mathbf{W_0}^j + \underbrace{\frac{1}{k}\sum_{i=1}^{k}(\mathbf{B}_i^j\mathbf{A}_i^j) - \frac{1}{k}\sum_{i=1}^{k}\mathbf{B}_i^j \times \frac{1}{k}\sum_{i=1}^{k}\mathbf{A}_i^j}_{\text{Residual}} \tag{6}$$

**DP-SGD.** DP-SGD (1) is a privacy-preserving variant of stochastic gradient descent (SGD) designed to ensure DP during training. It enforces privacy by clipping per-sample gradients to a fixed norm $C$ to limit their sensitivity and then adding isotropic Gaussian noise $\mathcal{N}(0, \sigma^2 C^2 \mathbf{I})$, where $\sigma$ controls the noise magnitude. The cumulative privacy loss over iterations is quantified using the moments accountant (51) and Rényi DP (38), which offer a tight bound on the final privacy parameter $\epsilon$.

**Exact Aggregation in Fed. LoRA: Tradeoff b/w Performance and Communication Costs.**

Standard federated averaging of individual LoRA adapters (FedIT (58)) introduces *inexactness* in aggregation, as the ideal update should be the average of client updates.

$$\underbrace{\mathbf{W}_0 + \frac{1}{c}\sum_{i=1}^{c}\mathbf{B}_i \times \frac{1}{c}\sum_{i=1}^{c}\mathbf{A}_i}_{\text{Vanilla aggregation in LoRA (FedIT)}} \neq \underbrace{\mathbf{W}_0 + \frac{1}{c}\sum_{i=1}^{c}(\mathbf{B}_i\mathbf{A}_i)}_{\text{Ideal aggregation}}. \tag{7}$$

The inexactness arises because the ideal averaged updates, given by $\sum_{i=1}^{c}\mathbf{B}_i\mathbf{A}_i$, often exceed rank $r$, violating the low-rank constraint imposed by LoRA. To address this, FedEx-LoRA and FLoRA introduce $\mathbf{W}_{\text{err}}$ as a higher-rank correction term within the pre-trained weight matrix $\mathbf{W}_0$, which is inherently high-rank. This correction ensures exact aggregation, leading to consistently improved performance over FedIT.

This, however, comes at the cost of increased communication. Since the error matrix is high rank, it substantially increases the amount of data transmitted per round. The communication cost is determined by the number of parameters sent during aggregation, which, for an $m \times n$ matrix, is proportional to its rank. As a result, in FedEx-LoRA and similar methods that enforce exact aggregation, communication cost scales linearly with the number of clients relative to Fed-IT. This becomes particularly concerning when the number of clients grows large, potentially requiring the transmission of the entire model's weights.

FFA-LoRA addresses inexact aggregation by keeping only $\mathbf{B}$ trainable while fixing $\mathbf{A}$ uniformly across clients. However, this comes at the cost of reduced expressivity and limits the benefits of jointly optimizing $\mathbf{A}$ and $\mathbf{B}$. As a result, performance degrades, as demonstrated previously (43). This stems from two factors: suboptimal individual updates and the need for higher-rank adaptations. Freezing $\mathbf{A}$ leads to suboptimal updates, even in centralized training, where FFA-LoRA underperforms compared to LoRA. Additionally, recent work (34) shows that models trained using FFA-LoRA progressively deviate from the optimal hypothesis. Empirical evidence shows that the advantages of exactness are outweighed by the degradation caused by these factors.

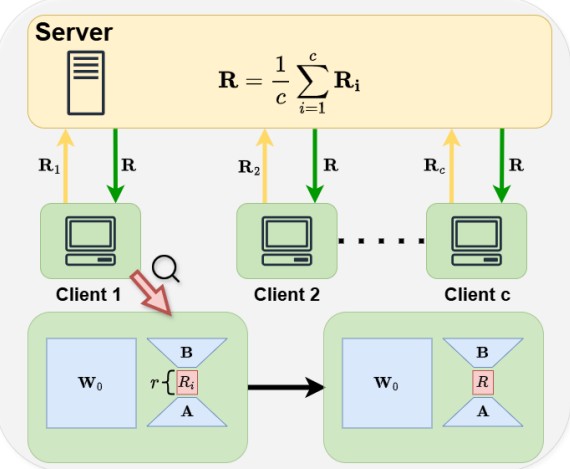

Figure 2: **Fed-SB**: Our method achieves optimal exact aggregation by averaging only the $r \times r$ matrices $\mathbf{R}_i$, significantly reducing communication costs.

**Private Fine-Tuning.** Pre-training on public data followed by FT on user-specific private data[*] is a common approach for adapting models under privacy constraints (54; 45). This two-stage process enhances performance in private learning while preserving user data privacy. FL naturally improves privacy by keeping data decentralized. However, even without direct data sharing, client-specific model updates can still leak sensitive information (50). Thus, developing privacy-preserving FT methods for FL is essential to ensure strong privacy guarantees while maintaining performance.

Training a model with DP-SGD introduces noise into the gradient, and consequently, into the model update itself. In the case of LoRA, this deviation from the ideal update is more pronounced than in full FT due to second-order noise terms. To illustrate this, let $\mathbf{A}$ and $\mathbf{B}$ represent the adapter updates learned without privacy. Under DP-SGD, these updates are perturbed by noise terms $\boldsymbol{\xi}_A$ and $\boldsymbol{\xi}_B$, respectively. The difference between the ideal update $\Delta\mathbf{W}$ and the noisy update $\Delta\mathbf{W}_{DP}$ is:

$$\Delta\mathbf{W}_{DP} - \Delta\mathbf{W} = (\mathbf{B} + \boldsymbol{\xi}_B)(\mathbf{A} + \boldsymbol{\xi}_A) - \mathbf{B}\mathbf{A} = \boldsymbol{\xi}_B\mathbf{A} + \mathbf{B}\boldsymbol{\xi}_A + \boldsymbol{\xi}_B\boldsymbol{\xi}_A. \tag{8}$$

The first-order noise term, $\boldsymbol{\xi}_B\mathbf{A} + \mathbf{B}\boldsymbol{\xi}_A$, is expected and occurs even in full FT with DP-SGD. However, the second-order noise term, $\boldsymbol{\xi}_B\boldsymbol{\xi}_A$, causes noise amplification, leading to further performance degradation in LoRA-based methods (44). This issue is exacerbated in FL, as individual client updates deviate even further from the ideal global update. FFA-LoRA avoids this problem by freezing $\mathbf{A}$, preventing the introduction of additional noise terms.

The bilinear parameterization in LoRA introduces two key challenges: inexact aggregation and noise amplification. FedEx-LoRA/FLoRA addresses the inexactness issue by enabling exact aggregation, but at the cost of communication overhead that scales prohibitively with the number of clients. FFA-LoRA mitigates inexact aggregation and excessive communication but sacrifices performance, as it operates in a low-rank space and has reduced expressivity. An ideal method would efficiently learn higher-rank updates while inherently enabling exact aggregation without increasing communication costs. However, any LoRA-based formulation that attempts to resolve these challenges must inevitably trade off expressivity, ultimately compromising performance. We prove that LoRA-SB provides an optimal reparameterization of the updates, effectively overcoming all limitations of LoRA in both non-private and privacy-preserving federated settings.

## 3 Method

**LoRA-SB for Fine-Tuning.** LoRA-SB (39) optimally approximates full FT gradients in low-rank spaces and demonstrates that its entire optimization trajectory aligns with the ideal low-rank projection of the full FT path. To achieve this, LoRA-SB fixes $\mathbf{A}$ and $\mathbf{B}$ while introducing a new trainable adapter $\mathbf{R}$ of size $r \times r$. Since $\mathbf{R}$ has rank $r$, it updates the pre-trained weight while maintaining rank $r$, making it highly parameter efficient. As a result, LoRA-SB consistently outperforms LoRA (and variants) across benchmarks while using 45–90x fewer trainable parameters.

**Fed-SB: A Silver bullet for (Private) Federated Fine-Tuning.** We propose **Fed-SB**, an extremely communication-efficient and high-performing federated adaptation of LoRA-SB. Instead of reparameterizing updates as a low-rank decomposition with learnable adapters, the server distributes frozen adapters $\mathbf{B}$ and $\mathbf{A}$, while clients train only a small matrix $\mathbf{R}$ (Figure 2). This enables exact aggregation, as the global update is simply the average of $\mathbf{R}$ across clients. Formally, given a pre-trained weight $\mathbf{W}_0$ and data distributed across $c$ clients, each client learns updates of the form:

$$\Delta \mathbf{W}_i = \mathbf{B}\mathbf{R}_i\mathbf{A}. \tag{9}$$

The server then aggregates the updates by computing the global $\mathbf{R}$ matrix:

$$\mathbf{R}^{\text{agg}} = \frac{1}{c}\sum_{i=1}^{c}\mathbf{R}_i, \Delta\mathbf{W}^{\text{agg}} = \mathbf{B}\left(\frac{1}{c}\sum_{i=1}^{c}\mathbf{R}_i\right)\mathbf{A}. \tag{10}$$

We show that **Fed-SB** effectively resolves all challenges in (private) federated FT while achieving state-of-the-art communication efficiency and performance. Table 1 highlights the advantages of Fed-SB over other methods. Since Fed-SB fixes the adapter matrices $A$ and $B$ throughout training, their initialization is crucial for effective learning. We adopt the update-based initialization strategy from LoRA-SB, which we detail in Appendix C.

**Fed-SB: Exact Aggregation.** Since only $\mathbf{R}$ is trainable, simple averaging of $\mathbf{R}$ across clients ensures exact aggregation without any updates to any other matrix. Further, the linearity of the global update with respect to the client-specific matrices $\mathbf{R}_i$ guarantees that exact aggregation occurs within rank $r$, preventing communication costs from scaling with number of clients. This is because the server only needs to aggregate and transmit $\mathbf{R}$, which can be proven by computing the global update $\Delta\mathbf{W}^{\text{agg}}$:

$$\Delta\mathbf{W}^{\text{agg}} = \mathbf{B}\left(\frac{1}{c}\sum_{i=1}^{c}\mathbf{R}_i\right)\mathbf{A}, \tag{11}$$

$$\Delta\mathbf{W}^{\text{agg}} = \frac{1}{c}\sum_{i=1}^{c}\mathbf{B}\mathbf{R}_i\mathbf{A} = \frac{1}{c}\sum_{i=1}^{c}\Delta\mathbf{W}_i. \tag{12}$$

Since the global update is simply the average of the individual updates, the aggregation is exact. The key advantage here is that this exact aggregation does not incur additional communication overhead like FedEx-LoRA, nor does it compromise individual update quality like FFA-LoRA.

**Fed-SB: Privacy.** Privacy-preserving FT with Fed-SB has two key advantages: 1) Fed-SB avoids noise amplification, which is a common issue in LoRA-based methods. 2) Since Fed-SB inherently requires fewer learnable parameters, the amount of noise added to enforce DP guarantees is significantly lower.

**Avoids Noise Amplification.** DP-SGD training in Fed-SB avoids second-order noise terms, as only $\mathbf{R}$ is trainable. This prevents the introduction of cross terms, thereby eliminating noise amplification. The difference between the updates with and without private training is given by:

$$\Delta\mathbf{W}_{DP} - \Delta\mathbf{W} = \mathbf{B}\left(\mathbf{R} + \boldsymbol{\xi}_B\right)\mathbf{A} - \mathbf{B}\mathbf{R}\mathbf{A} \implies \Delta\mathbf{W}_{DP} - \Delta\mathbf{W} = \mathbf{B}\boldsymbol{\xi}_B\mathbf{A}. \tag{13}$$

---

[*]Although pre-training data may be public, it often contains sensitive or proprietary information, raising privacy concerns. However, any privacy loss from pre-training has already occurred upon the model's release.

Since the private update remains linear in **R**, Fed-SB achieves the same benefits in private settings as FFA-LoRA, while avoiding its limitations.

**Fewer Learnable Parameters.** The noise added to gradients for DP enforcement increases with the number of trainable parameters ([4]; [1]; [9]), potentially distorting learning and degrading performance. Reducing trainable parameters improves DP performance, provided the model retains sufficient task-specific expressivity.

> **Lemma 1.** *Consider a model with $d$ learnable parameters trained using DP-SGD. The privacy parameter $\epsilon$ for $\delta$-approximate differential privacy, given $T$ training steps and a batch size of $q$, is expressed as:*
>
> $$\epsilon = O(q\sqrt{Td\log(1/\delta)}) = O(\sqrt{d}). \tag{14}$$
>
> *Proof.* See Appendix A. □

Lemma 1 establishes that reducing the number of learnable parameters enhances privacy guarantees under the same training setup. Specifically, achieving an equivalent level of privacy requires injecting less noise per parameter when fewer parameters are trained. Since LoRA-SB optimally approximates full fine-tuning gradients, its updates remain as effective as those in LoRA while benefiting from lower noise per update, resulting in a superior privacy-utility tradeoff. More generally, any reparameterization that reduces trainable parameters leads to a smaller accumulated privacy parameter $\epsilon$, thereby improving performance, provided the reduction does not compromise learning.

**Fed-SB.** Fed-SB has significantly less communication costs than other federated FT methods. This is due to two key reasons: 1) LoRA-SB achieves performance comparable to or better than LoRA while requiring 45-90x fewer trainable parameters. 2) Fed-SB aggregates only the $r \times r$ trainable matrix **R**, ensuring exact aggregation without additional communication overhead. This allows Fed-SB to leverage higher-rank updates without increasing communication costs. LoRA-SB typically operates at ranks 2-4x higher than LoRA, enabling Fed-SB to capture richer updates. Retaining high-rank information is crucial in FL ([34]) and a key factor in the superior performance of FedEx-LoRA/FLoRA over FFA-LoRA/Fed-IT beyond just aggregation exactness.

While our main focus is on the rank-homogeneous setting (where all clients use the same adapter rank), we also extend Fed-SB to support rank-heterogeneous clients, where each client trains with its own local rank budget. Additional details and results are provided in Table 9 (Appendix D), where we show that the rank-heterogeneous setup achieves performance comparable to the homogeneous rank settings.

## 4 Experiments & Results

Table 2: Federated fine-tuning of Llama-3.2 3B across eight commonsense reasoning datasets. # Comm. denotes the number of parameters communicated per round (in M). Best results are in **bold**.

| Method | Rank | # Comm. (↓) | Accuracy (↑) | | | | | | | | |
|---|---|---|---|---|---|---|---|---|---|---|---|
| | | | BoolQ | PIQA | SIQA | HellaS. | WinoG. | ARC-e | ARC-c | OBQA | Avg. |
| FedIT | 32 | 48.63 | 62.99 | 81.50 | 73.13 | 76.83 | 71.51 | 84.89 | 70.65 | 70.62 | 74.02 |
| FFA-LoRA | 32 | 24.31 | 62.87 | 80.03 | 68.53 | 70.02 | 65.56 | 82.95 | 66.38 | 66.85 | 70.40 |
| FedEx-LoRA | 32 | 243.15 | 65.05 | 82.81 | 74.67 | 81.84 | 76.01 | 86.32 | 71.42 | 73.81 | 76.49 |
| FLoRA | 32 | 243.15 | 65.05 | 82.81 | 74.67 | 81.84 | 76.01 | 86.32 | 71.42 | 73.81 | 76.49 |
| Fed-SB | 120 | 2.83 | 64.86 | 81.66 | 74.87 | 81.67 | 75.22 | 86.03 | 70.56 | 72.25 | 75.89 |
| Fed-SB | 160 | 5.02 | 65.57 | 82.37 | 76.15 | 84.10 | 77.98 | 86.62 | 72.10 | 73.63 | 77.32 |
| Fed-SB | 200 | 7.85 | **66.66** | **83.79** | **77.22** | **85.42** | **79.56** | **87.46** | **72.53** | **76.02** | **78.58** |

**Overview.** We evaluate across three diverse NLP benchmarks, covering models that span from BERT-base (110M) to Gemma-2 (9B), thereby encompassing both masked and autoregressive architectures. Specifically, we fine-tune Mistral-7B ([23]), Gemma-2 9B ([47]), Llama-3.2 3B ([16]), and BERT-base ([15]). Our experiments

Table 3: Federated fine-tuning of Llama-3.2 3B across eight commonsense reasoning datasets, in a highly data-heterogeneous setting, where each client is trained on a distinct dataset. # Comm. denotes the number of parameters communicated per round (in M). Best results are in **bold**.

| Method | Rank | # Comm. (↓) | Accuracy (↑) | | | | | | | | |
|--------|------|-------------|-------|------|------|--------|--------|-------|-------|------|------|
| | | | BoolQ | PIQA | SIQA | HellaS. | WinoG. | ARC-e | ARC-c | OBQA | Avg. |
| FedIT | 32 | 48.63 | 60.89 | 78.22 | 69.92 | 73.18 | 67.88 | 81.21 | 67.04 | 66.91 | 70.80 |
| FFA–LoRA | 32 | 24.31 | 60.73 | 76.91 | 65.37 | 65.18 | 61.89 | 79.41 | 62.92 | 63.12 | 67.17 |
| FedEx-LoRA | 32 | 243.15 | 62.55 | 79.36 | 71.41 | 78.12 | 72.45 | 82.89 | 67.88 | 70.25 | 73.13 |
| FLoRA | 32 | 243.15 | 62.55 | 79.36 | 71.41 | 78.12 | 72.45 | 82.89 | 67.88 | 70.25 | 73.13 |
| Fed-SB | 120 | 2.83 | 61.41 | 78.13 | 71.02 | 78.24 | 71.78 | 82.45 | 67.12 | 68.83 | 72.65 |
| Fed-SB | 160 | 5.02 | 62.34 | 79.05 | 72.39 | 80.52 | 74.67 | 83.18 | 68.64 | 70.12 | 73.98 |
| Fed-SB | 200 | 7.85 | **63.28** | **80.34** | **73.56** | **82.07** | **76.01** | **84.01** | **69.02** | **72.46** | **75.21** |

Table 4: Federated fine-tuning of Mistral-7B and Gemma-2 9B on GSM8K and MATH. # Comm. denotes the number of parameters communicated per round (in M). Best results are in **bold**.

| Model | Method | Rank | # Comm. (↓) | Accuracy (↑) | |
|-------|--------|------|-------------|-------|------|
| | | | | GSM8K | MATH |
| | FedIT | 32 | 83.88 | 52.91 | 12.26 |
| | FFA–LoRA | 32 | 41.94 | 53.67 | 12.46 |
| | FedEx-LoRA | 32 | 2097.34 | 54.28 | 12.92 |
| Mistral-7B | FLoRA | 32 | 2097.34 | 54.28 | 12.92 |
| | Fed-SB | 120 | 3.22 | 54.44 | **14.06** |
| | Fed-SB | 160 | 5.73 | 54.81 | 13.74 |
| | Fed-SB | 200 | 8.96 | **56.18** | 13.76 |
| | FedIT | 32 | 108.04 | 74.22 | 36.30 |
| | FFA–LoRA | 32 | 54.02 | 75.06 | 35.44 |
| | FedEx-LoRA | 32 | 2701.12 | 74.68 | 36.70 |
| Gemma-2 9B | FLoRA | 32 | 2701.12 | 74.68 | 36.70 |
| | Fed-SB | 120 | 4.23 | 74.75 | 36.36 |
| | Fed-SB | 160 | 7.53 | 76.88 | 36.94 |
| | Fed-SB | 200 | 11.76 | **77.03** | **37.56** |

consider both performance and communication efficiency. Detailed experimental and dataset specifications are provided in Appendix G and H, respectively. For federated data distribution, we adopt a standard protocol where client datasets are randomly sampled, following established practice in FL (44; 19; 29). We conduct experiments on a single NVIDIA A6000 GPU (48 GB) and report the average results from three independent runs.

**Baselines.** We evaluate against several SOTA federated FT approaches described previously, considering both private and non-private settings. Specifically, we compare it with **FedIT**, **FedEx-LoRA**, **FLoRA**, and **FFA-LoRA**. Where applicable, we also include comparisons with standard **LoRA** (21).

## 4.1 Instruction Tuning

**Details.** We conduct experiments in the federated non-private setting across two reasoning tasks: commonsense reasoning and arithmetic reasoning. For commonsense reasoning, we fine-tune Llama-3.2 3B on COMMONSENSE170K, a dataset aggregating eight commonsense reasoning corpora (22), and evaluate its effectiveness across all constituent datasets. The experiments are performed in a cross-silo federated learning setup involving 5 clients.

We also evaluate Fed-SB under extreme data heterogeneity. Instead of randomly sampling examples for each client, we assign each constituent dataset to a distinct client, resulting in a highly non-IID 8-client setup. Each client trains on a distinct distribution, with varying dataset sizes.

For arithmetic reasoning, we fine-tune Mistral-7B (23) and Gemma-2 9B (47) on 20K samples from the MetaMathQA dataset (55) and assess their performance on the GSM8K (13) and MATH (20) benchmarks. In this setup, we distribute the federated training across 25 clients. In both cases, we apply LoRA modules to the key, query, value, attention output, and all fully connected weights.

**Results** (Tables 2, 3, 4). Our method achieves state-of-the-art performance, outperforming all previous baselines in both accuracy and communication efficiency across all models and benchmarks. Figure 4 further illustrates this significant improvement. Additional results on the effect of varying rank are reported in Table 10 in Appendix E.

Table 5: Centralized (Cent.) private fine-tuning of BERT-base on SNLI for varying values of $\epsilon$. A smaller $\epsilon$ indicates a stricter privacy budget. # Params. denotes the number of trainable parameters (in K). Best results are in **bold**.

| Method | Rank | # Params. ($\downarrow$) | Accuracy ($\uparrow$) | | | | |
|---|---|---|---|---|---|---|---|
| | | | $\epsilon = 1$ | $\epsilon = 3$ | $\epsilon = 5$ | $\epsilon = 7.5$ | $\epsilon = 10$ |
| Cent. LoRA | 32 | 1181.96 | 66.49 | 67.79 | 68.17 | 70.78 | 70.81 |
| Cent. FFA-LoRA | 32 | 592.13 | 74.40 | 75.02 | 75.02 | 76.14 | 76.60 |
| Cent. Fed-SB | 32 | 26.88 | 73.99 | 75.09 | 74.45 | 77.01 | 76.24 |
| Cent. Fed-SB | 48 | 57.59 | **75.98** | 75.70 | 76.58 | 76.77 | 77.96 |
| Cent. Fed-SB | 64 | 100.61 | 75.81 | **77.07** | **77.59** | **78.75** | **78.08** |

Table 6: Federated private fine-tuning of BERT-base on SNLI for varying values of $\epsilon$. A smaller $\epsilon$ indicates a stricter privacy budget. # Comm. denotes the number of parameters communicated per round (in K). Best results are in **bold**.

| Method | Rank | # Comm. ($\downarrow$) | Accuracy ($\uparrow$) | | | | |
|---|---|---|---|---|---|---|---|
| | | | $\epsilon = 1$ | $\epsilon = 3$ | $\epsilon = 5$ | $\epsilon = 7.5$ | $\epsilon = 10$ |
| FedIT | 32 | 1181.96 | 49.57 | 51.29 | 48.53 | 55.63 | 60.96 |
| FFA-LoRA | 32 | 592.13 | 70.11 | 71.49 | 72.69 | 73.27 | 74.02 |
| FedEx-LoRA | 32 | 3541.26 | 67.38 | 69.68 | 72.92 | 71.89 | 74.33 |
| FLoRA | 32 | 3541.26 | 67.38 | 69.68 | 72.92 | 71.89 | 74.33 |
| Fed-SB | 32 | 26.88 | 70.33 | 72.68 | 73.57 | 73.62 | 73.85 |
| Fed-SB | 48 | 57.59 | 73.7 | 74.74 | 73.66 | 74.75 | 75.02 |
| Fed-SB | 64 | 100.61 | **73.83** | **74.88** | **76.27** | **75.75** | **75.86** |

**Commonsense Reasoning** (Table 2). Fed-SB ($r = 200$) achieves an average improvement of 4.56% over FedIT while requiring 6× lower communication cost. Additionally, Fed-SB ($r = 200$) surpasses the previous SOTA performance methods FedEx-LoRA/FLoRA by 2.09%, while reducing communication cost by an impressive 31×. Notably, while the communication cost of FedEx-LoRA/FLoRA scales linearly with the number of clients, our method maintains a constant, client-independent communication cost. These results are obtained with just 5 clients, implying that the full extent of our method's communication efficiency is not fully depicted here. As the number of clients increases, the relative advantage of Fed-SB over existing methods grows even further.

**Highly Data-Heterogenous Setting** (Table 3). Fed-SB significantly outperforms all other methods even in this highly non-IID setting. Specifically, Fed-SB ($r = 200$) surpasses the previous state-of-the-art methods, FedEx-LoRA and FLoRA, by 2.08% in accuracy while achieving a remarkable 31× reduction in communication cost.

**Arithmetic Reasoning** (Table 4). For Mistral-7B, Fed-SB ($r = 200$) outperforms FedEx-LoRA/FLoRA on GSM8K by 1.90%, while achieving an impressive 234× reduction in communication cost. Additionally,

Fed-SB ($r = 200$) surpasses FFA-LoRA on GSM8K by 2.51%, with approximately 5× lower communication cost. For Gemma-2 9B, Fed-SB ($r = 200$) outperforms FedEx-LoRA/FLoRA on MATH by 0.86%, while reducing communication cost by 230×.

## 4.2 (Federated) Private Fine-Tuning

**Details.** We fine-tune BERT-base (15) on SNLI (8), a standard benchmark for natural language inference. Following LoRA(21), we apply LoRA modules only to the self-attention layers. Our evaluation considers two DP settings: a centralized private setup and a federated private setup. To enforce DP guarantees during training, we use the Opacus library (53) with the DP-SGD optimizer (1). In the federated setting, training is conducted in a cross-silo setup with 3 clients. We conduct experiments across a range of privacy budgets, varying $\epsilon$ from 1 to 10.

**Results** (Tables 5, 6). Fed-SB consistently outperforms all prior baselines in both accuracy and communication/parameter efficiency across all privacy budgets in both settings. Figures 5, 6, and 7 further illustrate this significant improvement. Further experiments analyzing the impact of rank variation are given in Table 11 (Appendix E).

**Centralized Private** (Table 5). Fed-SB showcases significant improvement over other methods while using only a fraction of the parameters, across all $\epsilon$ values. For instance, at $\epsilon = 3$, Fed-SB ($r = 64$) surpasses centralized LoRA and centralized FFA-LoRA by 9.28% and 2.05%, respectively, while using ≈ 12x and 6x fewer parameters.

**Federated Private** (Table 6). Fed-SB consistently outperforms all previous methods across all values of $\epsilon$, while significantly reducing communication costs. For instance, at $\epsilon = 1$, Fed-SB ($r = 64$) outperforms FedIT, FedEx-LoRA/FLoRA, and FFA-LoRA by 24.26%, 6.48%, and 2.72%, respectively, while reducing communication cost by approximately 12x, 35x, and 6x. FedIT performs significantly worse in the federated private setting compared to the federated non-private setting. We hypothesize that this is due to increased deviation in updates under DP constraints and added noise, leading to greater divergence from the ideal.

## 4.3 Memory and Training Time

**Memory.** Fed-SB needs lower per-client training memory relative to all other baselines by substantially reducing the number of trainable parameters. Notably, this advantage holds even when Fed-SB is trained with a higher rank ($r = 200$), where it still requires less memory than competing methods at a lower rank ($r = 32$). We note that the peak memory usage of Fed-SB never exceeds that of any other federated LoRA-based baseline. We benchmark the peak per-client training memory for all models and configurations used in our study in Table 7.

Table 7: Peak per-client training memory (in GB) for different methods across the various models used in this work. Fed-SB consistently exhibits lower memory usage across all model configurations.

| Method | Rank | Peak Memory (GB) | | |
|---|---|---|---|---|
| | | **Mistral-7B** | **Gemma-2 9B** | **Llama-3.2 3B** |
| FedIT | 32 | 15.92 | 19.99 | 7.71 |
| FFA-LoRA | 32 | 15.51 | 19.44 | 7.46 |
| FedEx-LoRA | 32 | 15.92 | 19.99 | 7.71 |
| FLoRA | 32 | 15.92 | 19.99 | 7.71 |
| Fed-SB | 200 | 15.18 | 19.03 | 7.30 |

**Training Time.** Fed-SB introduces a negligible training time overhead compared to other methods, primarily due to its lightweight initialization process. To quantify this, we measure the additional training time introduced by Fed-SB relative to the average per-epoch training time per client in baseline methods. These measurements are conducted across the various experimental settings described in our paper. As shown in Table 8, the overhead remains consistently minimal, approximately 2%, across multiple model configurations.

Table 8: Training time overhead introduced by Fed-SB ($r = 200$) relative to the average per-epoch training time per client in baseline methods. The overhead is minimal ($\approx 2\%$) across different model configurations.

| Model | Fed-SB Overhead (mm:ss) | Avg. Epoch Time / Client (mm:ss) |
|---|---|---|
| Mistral-7B | 00:13 | 09:22 |
| Gemma-2 9B | 00:16 | 12:43 |
| Llama-3.2 3B | 01:43 | 62:54 |

## 5  Conclusion

Existing LoRA-based federated FT methods either suffer from suboptimal updates or incur prohibitively high communication costs. We introduce Fed-SB, a federated adaptation of LoRA-SB that ensures exact aggregation while maintaining high communication efficiency. By training only a small $r \times r$ matrix and leveraging direct averaging, Fed-SB eliminates high-rank update costs and achieves communication efficiency independent of the number of clients. Fed-SB is particularly well-suited for private FT, as its linearity prevents noise amplification, and its reduced parameter count minimizes noise required for enforcing DP guarantees. It consistently achieves a new state-of-the-art across all models and tasks while reducing communication costs by up to 230x. These advantages establish Fed-SB as an efficient and scalable solution for (private) federated FT.

## 6  Limitations

While our approach is easily adaptable to other architectures, such as VLMs and VITs, we have not evaluated on such architectures, or on vision/multimodal settings. Given the strong performance of our method in non-IID settings, it also holds promise for adapter merging, where adapters are independently trained on distinct tasks, a scenario we have not yet evaluated.

## 7  Acknowledgements

This work was supported by funding from Mohamed bin Zayed University of Artificial Intelligence (MBZUAI) and an ADIA Lab fellowship.

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

# Appendix

## Contents

## A   Proof of Lemma 1

> *Lemma.* Consider a model with $d$ learnable parameters trained using DP-SGD. The privacy parameter $\epsilon$ for $\delta$-approximate differential privacy, given $T$ training steps and a batch size of $q$, is expressed as:
>
> $$\epsilon = O(q\sqrt{Td\log(1/\delta)}) = O(\sqrt{d}). \tag{15}$$

*Proof.* The following result (1) describes the relationship between noise variance, privacy parameters, number of optimization steps, batch size, and sample size in DP-SGD.

*Theorem.* There exist constants $c_1$ and $c_2$ such that, given the sampling probability $q = L/N$ and the number of optimization steps $T$, for any $\epsilon < c_1 q^2 T$, DP-SGD is $(\epsilon, \delta)$-differentially private for any $\delta > 0$ if the noise scale satisfies:

$$\sigma \geq c_2 \frac{q\sqrt{T\log(1/\delta)}}{\epsilon}. \tag{16}$$

Each DP-SGD step introduces noise following $\mathcal{N}\left(0, \sigma^2 C^2 \mathbf{I}_d\right)$ and satisfies $(\alpha, \alpha/(2\sigma^2))$-RDP (Rényi DP) for the Gaussian mechanism. For a function with $\ell_2$-sensitivity $\Delta_2$, the Gaussian mechanism satisfies $(\alpha, \epsilon)$-RDP with:

$$\epsilon(\alpha) = \frac{\alpha \Delta_2^2}{2\sigma_{\text{noise}}^2}. \tag{17}$$

Since DP-SGD has $\Delta_2 = C$ and $\sigma_{\text{noise}} = \sigma C$, applying privacy amplification due to sampling probability $q$ results in each step satisfying $(\alpha, \gamma)$-RDP, where, for small $q$:

$$\gamma = O\left(\frac{q^2\alpha}{\sigma^2}\right). \tag{18}$$

Using composition over $T$ steps, the total RDP privacy parameter becomes:

$$\gamma_{\text{total}} = O\left(\frac{q^2 T\alpha}{\sigma^2}\right). \tag{19}$$

Converting this RDP bound back to $(\epsilon, \delta)$-DP and setting $\alpha$ proportional to $1/\sqrt{d}$, given that the $\ell_2$-norm of the gradient scales as $\sqrt{d}$, we obtain:

$$\epsilon = O\left(\frac{q^2 T\alpha}{\sigma^2} + \frac{\log(1/\delta)}{\alpha - 1}\right). \tag{20}$$

Substituting $\sigma \propto 1/\sqrt{d}$, we derive:

$$\epsilon = O(q\sqrt{Td\log(1/\delta)}) = O(\sqrt{d}). \tag{21}$$

$\square$

## B  Related Work

**Parameter-Efficient Fine-Tuning (PEFT).** LoRA (21) has become ubiquitous for fine-tuning LLMs (57) by modeling weight updates as product of low-rank matrices. Several variants have been proposed to improve efficiency, stability, and adaptability. QLoRA (14) enables efficient fine-tuning through quantization strategies, reducing memory usage while maintaining performance. AdaLoRA (60) dynamically allocates a layer-specific rank budget by assigning importance scores to individual weight matrices. LoRA-XS (5) further reduces trainable parameters by inserting a trainable matrix between frozen LoRA matrices. VeRA (27) enhances parameter efficiency by learning shared adapters across layers. DoRA (32) decomposes the pre-trained matrix into two parts—*magnitude* and *direction*—and applies LoRA modules only to the *direction* component. PiSSA (36) improves adaptation by initializing adapters using the singular value decomposition (SVD) of pre-trained weights. rsLoRA (25) introduces a rank-scaling factor to stabilize learning. LoRA-SB (39) provably approximates gradients optimally in low-rank spaces, achieving superior performance with significantly higher parameter efficiency.

**Federated Fine-Tuning.** Federated Learning (FL) consists of a centralized global model and multiple clients, each with its own local dataset and computational capacity. The global model is updated by aggregating client updates (24). FedBERT (48) focuses on federated pre-training, while other methods work on federated fine-tuning (61; 28; 3). Fed-IT (59) aggregates low-rank adapters across clients using standard federated averaging (35) before updating the global model. To address inexact aggregation, FedEx-LoRA (43) introduces an error matrix to correct residual errors, ensuring more precise updates. FLoRA (52) follows the same exact aggregation principle by stacking matrices and extends this approach to heterogeneous rank settings. FFA-LoRA (44) mitigates aggregation inexactness by freezing $\mathbf{A}$ and updating only the trainable low-rank adapter, averaging the latter to compute the global update. In some scenarios, clients require heterogeneous LoRA ranks due to varying computational budgets (63; 30). Methods like HetLoRA (10) enable rank heterogeneity through self-pruning and sparsity-aware aggregation strategies, but incur significant overhead. Beyond data heterogeneity, FL systems must also address Byzantine attacks, where malicious clients send corrupted updates. RAGA (64) propose a geometric median-based aggregation method that provides convergence guarantees under both Byzantine attacks and data heterogeneity for non-convex objectives, offering complementary robustness considerations to the privacy-focused approaches in Fed-SB.

**Differential Privacy (DP) and FL.** A common limitation of standard FL frameworks is their susceptibility to privacy attacks, as clients publicly share model updates with a central server. To address this issue, DP is incorporated into FL methods to ensure the privacy of client updates. This work follows the approximate DP framework (17; 18), which provides formal privacy guarantees for model updates. Privacy is enforced during training using the DP-SGD optimizer (1), which applies gradient clipping and noise injection to protect individual contributions.

Prior work (62) demonstrates that Huber loss minimization can improve mean estimation under user-level DP by reducing bias from clipping operations, particularly for heavy-tailed distributions and imbalanced user contributions. This approach complements the DP-SGD framework used in Fed-SB by addressing challenges in robust aggregation under privacy constraints. Since DP is preserved under composition and post-processing (17; 31), the final global model update also retains DP guarantees. Prior methods, such as Fed-IT and FedEx-LoRA, did not explicitly incorporate DP. This study extends these approaches to DP settings and benchmarks them alongside FFA-LoRA and the proposed method.

## C  Initialization in Fed-SB

Fed-SB adopts the initialization strategy introduced in LoRA-SB to fix the adapter matrices $B$ and $A$. Proper initialization is crucial, since $B$ and $A$ remain frozen during training. For instance, if $B$ were initialized to zero (as in standard LoRA), the product $BRA$ would remain zero throughout, preventing any learning. In contrast, initializing $B$ and $A$ as orthonormal matrices ensures well-scaled gradients and allows Fed-SB to nearly match the performance of full fine-tuning.

To construct $B$ and $A$, we approximate the optimal update by averaging the first-step update across a small set of samples. A truncated SVD of this estimated update is then used to initialize the adapters. This requires only a small fraction of the training data (typically 0.1%), leading to negligible overhead in computation and time. Since the update is computed layerwise, memory usage during initialization never exceeds that of subsequent Fed-SB fine-tuning and remains below that of LoRA. Empirical analysis in LoRA-SB (39) shows that even 0.1% of the samples is sufficient for stable initialization.

## D  Extensions to Rank-Heterogeneous Setting

In real-world federated deployments, client devices often operate under diverse computational budgets and memory constraints. This naturally leads to *rank-heterogeneous settings*, where different clients cannot train adapters of the same rank. Supporting such heterogeneity is important for practical adoption: while high-resource clients can benefit from richer low-rank subspaces, low-resource clients should still be able to participate meaningfully without being excluded from collaboration.

### D.1  Rank-Heterogeneous Fed-SB

We extend Fed-SB to explicitly handle rank-heterogeneous clients while preserving its guarantees of exact aggregation. The key idea is to align all clients in a shared basis, chosen as the top $r_{\max}$ singular vectors of a reference weight matrix. Each client $i$ then selects a local rank budget $r_i \leq r_{\max}$ and optimizes within its most informative subspace:

$$A_i = A[:, : r_i], \quad B_i = B[: r_i, :], \quad R_i = R[: r_i, : r_i].$$

During aggregation, each client's update $R_i$ is zero-padded (along rows and columns) to match the global dimension $r_{\max} \times r_{\max}$. This ensures that all updates are aligned in the same coordinate system and can be averaged exactly:

$$R_{\mathrm{agg}} = \frac{1}{c} \sum_{i=1}^{c} \mathrm{pad}(R_i), \quad \Delta W = B R_{\mathrm{agg}} A.$$

In this formulation, low-rank clients contribute updates restricted to their subspaces, while high-rank clients provide richer information, and all updates combine seamlessly. Thus, Fed-SB can support heterogeneous client capabilities without loss of information, while maintaining exactness of aggregation.

## D.2 Experiments

Table 9: Comparison of homogeneous and heterogeneous Fed-SB configurations for federated fine-tuning of Llama-3.2 3B on eight commonsense reasoning datasets.

| Method | BoolQ | PIQA | SIQA | HellaS. | WinoG. | ARC-e | ARC-c | OBQA | Avg. |
|---|---|---|---|---|---|---|---|---|---|
| Homogeneous (all ranks = 120) | 64.86 | 81.66 | 74.87 | 81.67 | 75.22 | 86.03 | 70.56 | 72.25 | 75.89 |
| Heterogeneous (effective rank = 120) | 64.34 | 81.50 | 74.23 | 81.02 | 74.88 | 85.89 | 70.65 | 71.62 | 75.52 |
| Homogeneous (all ranks = 160) | 65.57 | 82.37 | 76.15 | 84.10 | 77.98 | 86.62 | 72.10 | 73.63 | 77.32 |
| Heterogeneous (effective rank = 160) | 64.83 | 82.05 | 76.43 | 83.92 | 77.53 | 85.96 | 71.90 | 72.98 | 76.95 |

To assess the effectiveness of our federated rank-heterogeneous approach, we extend the commonsense reasoning experiments with Llama-3.2 3B to heterogeneous rank settings. For a fair comparison, we match the total rank budget of the homogeneous baselines ($120^2$ and $160^2$) by assigning client-specific ranks of $\{40, 40, 120, 120, 200\}$ and $\{60, 60, 180, 200, 220\}$, respectively. As shown in Table 9, Fed-SB achieves performance comparable to its homogeneous counterparts in both cases, demonstrating strong robustness to rank heterogeneity.

# E Effect of Varying Rank on Fed-SB Performance

To further investigate the role of the rank parameter $r$, we conduct ablation studies of Fed-SB in both standard federated and privacy-preserving settings. In the non-private setting, we evaluate Mistral-7B and Gemma-2 9B fine-tuned on a subset of MetaMathQA across a wide range of rank values ($r = 32$–$240$), with results reported in Table 10. While selecting an optimal rank remains an open problem for all LoRA-based methods, our experiments show that intermediate values ($r = 120$–$200$) generally offer the best trade-off between performance and efficiency.

In the privacy-preserving setting, we evaluate centralized private Fed-SB using BERT-base fine-tuned on SNLI across ranks ranging from 16 to 80, with results presented in Table 11. Here, we observe that ranks in the range of 48–80 consistently achieve the strongest performance across different privacy budgets.

Overall, owing to Fed-SB's lightweight design, we can scale to higher ranks when resources allow, yielding further performance improvements without incurring memory bottlenecks.

Table 10: Effect of varying Fed-SB rank ($r$) on federated fine-tuning performance of Mistral-7B and Gemma-2 9B, evaluated on GSM8K and MATH. Best results are in **bold**.

| Rank | Mistral-7B | | Gemma-2 9B | |
|---|---|---|---|---|
| | GSM8K (↑) | MATH (↑) | GSM8K (↑) | MATH (↑) |
| 32 | 53.76 | 12.88 | 73.78 | 35.92 |
| 64 | 53.93 | 13.31 | 74.32 | 36.05 |
| 96 | 54.38 | 13.56 | 74.66 | 36.23 |
| 120 | 54.44 | **14.06** | 74.75 | 36.36 |
| 160 | 54.81 | 13.74 | 76.88 | 36.94 |
| 200 | 56.18 | 13.76 | 77.03 | **37.56** |
| 240 | **56.32** | 13.74 | **77.14** | 37.34 |

# F Loss Curves

We present the training loss curves for the different methods in Figure 3 on Llama-3.2 3B for commonsense reasoning. All methods exhibit stable convergence, indicating that they are trained effectively.

Table 11: Effect of varying Fed-SB rank ($r$) on centralized private fine-tuning performance of BERT-base, evaluated on SNLI, under various privacy budgets ($\epsilon$). A smaller $\epsilon$ indicates a stricter privacy budget. Best results are in **bold**.

| Rank | Accuracy ($\uparrow$) | | | | |
|---|---|---|---|---|---|
| | $\epsilon = 1$ | $\epsilon = 3$ | $\epsilon = 5$ | $\epsilon = 7.5$ | $\epsilon = 10$ |
| 16 | 73.26 | 74.21 | 73.68 | 76.23 | 75.80 |
| 24 | 73.65 | 74.78 | 73.92 | 76.88 | 76.02 |
| 32 | 73.99 | 75.09 | 74.45 | 77.01 | 76.24 |
| 48 | **75.98** | 75.70 | 76.58 | 76.77 | 77.96 |
| 64 | 75.81 | **77.07** | **77.59** | 78.75 | 78.08 |
| 80 | 75.93 | 76.87 | 77.35 | **78.81** | **78.23** |

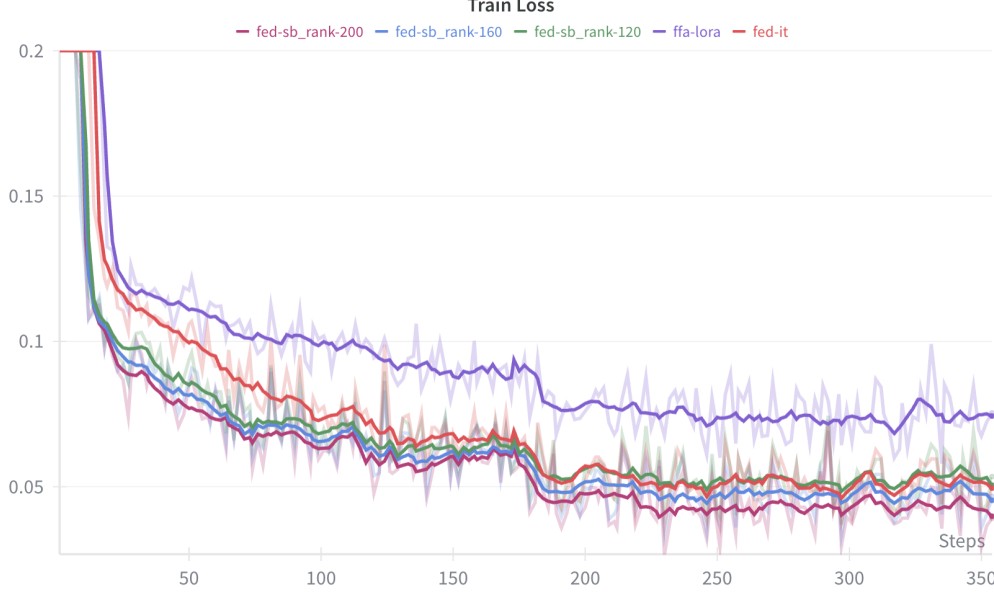

Figure 3: Training loss curves for various methods for a representative single client on Llama-3.2 3B for commonsense reasoning.

## G  Experiment Details

We conduct experiments on a single NVIDIA A6000 GPU (48 GB) and report the average results from three independent runs. All non-private models are trained using the AdamW optimizer (33). To optimize memory efficiency, all base models (except BERT) are loaded in `torch.bfloat16`. In line with LoRA-SB (39), we initialize the adapter matrices using just 1/1000 (0.1%) of the respective training dataset size.

**Instruction Tuning.** Table 12 presents the key hyperparameters and configurations for Mistral-7B, Gemma-2 9B, and Llama-3.2 3B. Our setup closely follows previous works (22; 39), ensuring consistency with established best practices. For the baseline experiments, we further set $\alpha = 16$, consistent with prior literature (43; 44). We additionally perform a sweep over the learning rate for our experiments.

**(Federated) Private Fine-Tuning.** Table 13 outlines the key hyperparameters and configurations for BERT-base in both centralized private and federated private settings. We train our models using the Opacus library (53) with the DP-SGD optimizer (1). Following standard DP practices, we set the privacy parameter as $\delta = \frac{1}{|\text{trainset}|}$. To ensure adherence to best practices, we adopt hyperparameter choices from prior works

(43; 21). For baseline experiments, we additionally set $\alpha = 16$, aligning with previous literature (43; 44). We additionally perform a sweep over the learning rate and maximum gradient norm in DP-SGD for our experiments.

Table 12: Hyperparameter settings for Mistral-7B, Gemma-2 9B, and Llama-3.2 3B.

|  | **Mistral-7B** | **Gemma-2 9B** | **Llama-3.2 3B** |
| --- | --- | --- | --- |
| Optimizer | AdamW | AdamW | AdamW |
| Learning Rate | 5e−4 | 5e−4 | 2e−4 |
| LR Scheduler | Cosine | Cosine | Linear |
| Warmup Ratio | 0.02 | 0.02 | 0.02 |
| Batch Size | 1 | 1 | 8 |
| Grad Acc. Steps | 32 | 32 | 24 |
| Max. Seq. Len | 512 | 512 | 256 |
| Dropout | 0 | 0 | 0 |
| # Clients | 25 | 25 | 5 |
| Local Epochs | 1 | 2 | 2 |
| Rounds | 1 | 1 | 1 |

Table 13: Hyperparameter settings for BERT-base in centralized private and federated private setups.

|  | **BERT-base (centralized)** | **BERT-base (federated)** |
| --- | --- | --- |
| Optimizer | DP-SGD | DP-SGD |
| Learning Rate | 5e−4 | 5e−4 |
| LR Scheduler | - | - |
| Warmup Ratio | 0 | 0 |
| Batch Size | 32 | 32 |
| Max. Phy. Batch Size | 8 | 8 |
| Max. Seq. Len | 128 | 128 |
| Dropout | 0.05 | 0.05 |
| Max. Grad. Norm | 0.1 | 0.1 |
| Epochs | 3 | - |
| # Clients | - | 3 |
| Local Epochs | - | 6 |
| Rounds | - | 1 |

## H  Dataset Details

**CommonSense170K** is a large-scale dataset that brings together eight benchmarks designed to assess various aspects of commonsense reasoning (22). Below is an overview of its constituent datasets:

1. **PIQA** (6) evaluates physical commonsense by asking models to determine the most reasonable action in a given scenario.

2. **ARC Easy (ARC-e)** (12) consists of elementary-level science questions, serving as a fundamental test of a model's reasoning abilities.

3. **OBQA** (37) presents knowledge-intensive, open-book multiple-choice questions that require multi-step reasoning and retrieval.

4. **HellaSwag** (56) tests contextual reasoning by asking models to predict the most plausible continuation of a passage from a set of candidates.

5. **SIQA** (42) examines social intelligence, requiring models to predict human actions and their social consequences.

6. **ARC Challenge (ARC-c)** (12) includes difficult multiple-choice science questions that demand deeper logical inference beyond statistical co-occurrence.

7. **BoolQ** (11) consists of naturally occurring yes/no questions, requiring models to infer relevant information from provided contexts.

8. **WinoGrande** (41) assesses commonsense knowledge through binary-choice sentence completion tasks that require resolving ambiguities.

The **MetaMathQA** dataset (55) constructs mathematical questions by reformulating them from different viewpoints while preserving their original knowledge content. We assess its performance using two well-established benchmarks: (1) **GSM8K** (13), a collection of grade-school-level math problems requiring step-by-step reasoning to reach a solution, and (2) **MATH** (20), which consists of high-difficulty, competition-style problems designed to test advanced mathematical skills.

**Stanford Natural Language Inference (SNLI)** is a widely used benchmark for assessing textual entailment models in natural language understanding. It contains approximately 570,000 sentence pairs, each categorized into one of three classes: entailment, contradiction, or neutral, requiring models to infer the relationship between a given premise and hypothesis.

## I Additional Results

We present the standard deviation results across multiple runs in Tables 14, 15, 16, 18, and 17.

Table 14: Standard deviation results on Llama-3.2 3B across commonsense reasoning benchmarks.

| Method | Rank | # Comm. | Std. Dev. (Accuracy) | | | | | | | | |
| --- | --- | --- | --- | --- | --- | --- | --- | --- | --- | --- | --- |
| | | | BoolQ | PIQA | SIQA | HellaS. | WinoG. | ARC-e | ARC-c | OBQA | Avg. |
| FedIT | 32 | 48.63 | 0.83 | 0.91 | 1.18 | 0.97 | 1.21 | 0.74 | 1.09 | 0.88 | 0.98 |
| FFA-LoRA | 32 | 24.31 | 0.95 | 0.79 | 1.24 | 1.05 | 1.13 | 0.86 | 1.17 | 0.93 | 1.01 |
| FedEx-LoRA | 32 | 243.15 | 0.88 | 0.77 | 1.02 | 1.01 | 0.94 | 0.73 | 1.08 | 0.80 | 0.90 |
| FLoRA | 32 | 243.15 | 0.76 | 0.88 | 1.07 | 1.02 | 0.99 | 0.69 | 1.04 | 0.90 | 0.92 |
| Fed-SB | 120 | 2.83 | 0.71 | 0.85 | 1.10 | 0.92 | 1.05 | 0.68 | 1.00 | 0.84 | 0.89 |
| Fed-SB | 160 | 5.02 | 0.89 | 0.72 | 1.15 | 0.94 | 1.08 | 0.66 | 0.97 | 1.01 | 0.93 |
| Fed-SB | 200 | 7.85 | 0.79 | 0.93 | 0.96 | 0.89 | 1.12 | 0.65 | 0.92 | 0.98 | 0.90 |

Table 15: Standard deviation results on Llama-3.2 3B across commonsense reasoning benchmarks under a highly data-heterogenous setting.

| Method | Rank | # Comm. | Std. Dev. (Accuracy) | | | | | | | | |
| --- | --- | --- | --- | --- | --- | --- | --- | --- | --- | --- | --- |
| | | | BoolQ | PIQA | SIQA | HellaS. | WinoG. | ARC-e | ARC-c | OBQA | Avg. |
| FedIT | 32 | 48.63 | 0.88 | 0.97 | 1.21 | 0.93 | 1.18 | 0.72 | 1.05 | 0.90 | 0.98 |
| FFA-LoRA | 32 | 24.31 | 1.02 | 0.84 | 1.26 | 1.09 | 1.11 | 0.88 | 1.14 | 0.95 | 1.04 |
| FedEx-LoRA | 32 | 243.15 | 0.90 | 0.81 | 1.03 | 1.02 | 0.93 | 0.75 | 1.07 | 0.83 | 0.92 |
| FLoRA | 32 | 243.15 | 0.81 | 0.91 | 1.08 | 1.03 | 0.96 | 0.69 | 1.01 | 0.87 | 0.92 |
| Fed-SB | 120 | 2.83 | 0.74 | 0.89 | 1.12 | 0.91 | 1.04 | 0.70 | 0.98 | 0.86 | 0.91 |
| Fed-SB | 160 | 5.02 | 0.93 | 0.76 | 1.17 | 0.95 | 1.09 | 0.66 | 0.94 | 1.00 | 0.94 |
| Fed-SB | 200 | 7.85 | 0.82 | 0.96 | 0.97 | 0.88 | 1.15 | 0.68 | 0.90 | 0.99 | 0.92 |

## J Additional Plots

Table 16: Standard deviation results on Mistral-7B and Gemma-2 9B across arithmetic reasoning benchmarks.

| Model | Method | Rank | # Comm. | Std. Dev. (Accuracy) | |
|---|---|---|---|---|---|
| | | | | GSM8K | MATH |
| Mistral-7B | FedIT | 32 | 83.88 | 0.96 | 1.18 |
| | FFA-LoRA | 32 | 41.94 | 0.82 | 1.25 |
| | FedEx-LoRA | 32 | 2097.34 | 1.07 | 1.02 |
| | FLoRA | 32 | 2097.34 | 0.91 | 1.11 |
| | Fed-SB | 120 | 3.22 | 0.77 | 1.05 |
| | Fed-SB | 160 | 5.73 | 0.89 | 0.94 |
| | Fed-SB | 200 | 8.96 | 0.74 | 1.00 |
| Gemma-2 9B | FedIT | 32 | 108.04 | 1.03 | 1.16 |
| | FFA-LoRA | 32 | 54.02 | 0.89 | 1.30 |
| | FedEx-LoRA | 32 | 2701.12 | 0.83 | 1.08 |
| | FLoRA | 32 | 2701.12 | 0.99 | 1.17 |
| | Fed-SB | 120 | 4.23 | 0.86 | 1.02 |
| | Fed-SB | 160 | 7.53 | 0.79 | 1.11 |
| | Fed-SB | 200 | 11.76 | 0.93 | 0.99 |

Table 17: Standard deviation results for centralized private fine-tuning on SNLI.

| Method | Rank | # Params. | Std. Dev. (Accuracy) | | | | |
|---|---|---|---|---|---|---|---|
| | | | $\epsilon = 1$ | $\epsilon = 3$ | $\epsilon = 5$ | $\epsilon = 7.5$ | $\epsilon = 10$ |
| Cent. LoRA | 32 | 1181.96 | 0.78 | 0.71 | 0.69 | 0.63 | 0.61 |
| Cent. FFA-LoRA | 32 | 592.13 | 0.66 | 0.60 | 0.58 | 0.54 | 0.52 |
| Cent. Fed-SB | 32 | 26.88 | 0.64 | 0.59 | 0.61 | 0.53 | 0.55 |
| Cent. Fed-SB | 48 | 57.59 | 0.59 | 0.57 | 0.54 | 0.52 | 0.50 |
| Cent. Fed-SB | 64 | 100.61 | 0.57 | 0.52 | 0.50 | 0.47 | 0.48 |

Table 18: Standard deviation results for federated private fine-tuning on SNLI.

| Method | Rank | # Comm. | Std. Dev. (Accuracy) | | | | |
|---|---|---|---|---|---|---|---|
| | | | $\epsilon = 1$ | $\epsilon = 3$ | $\epsilon = 5$ | $\epsilon = 7.5$ | $\epsilon = 10$ |
| FedIT | 32 | 1181.96 | 0.98 | 0.91 | 0.95 | 0.88 | 0.82 |
| FFA-LoRA | 32 | 592.13 | 0.86 | 0.80 | 0.78 | 0.75 | 0.73 |
| FedEx-LoRA | 32 | 3541.26 | 0.90 | 0.84 | 0.79 | 0.82 | 0.76 |
| FLoRA | 32 | 3541.26 | 0.89 | 0.83 | 0.78 | 0.81 | 0.75 |
| Fed-SB | 32 | 26.88 | 0.84 | 0.79 | 0.76 | 0.74 | 0.72 |
| Fed-SB | 48 | 57.59 | 0.80 | 0.76 | 0.74 | 0.71 | 0.70 |
| Fed-SB | 64 | 100.61 | 0.78 | 0.74 | 0.71 | 0.69 | 0.68 |

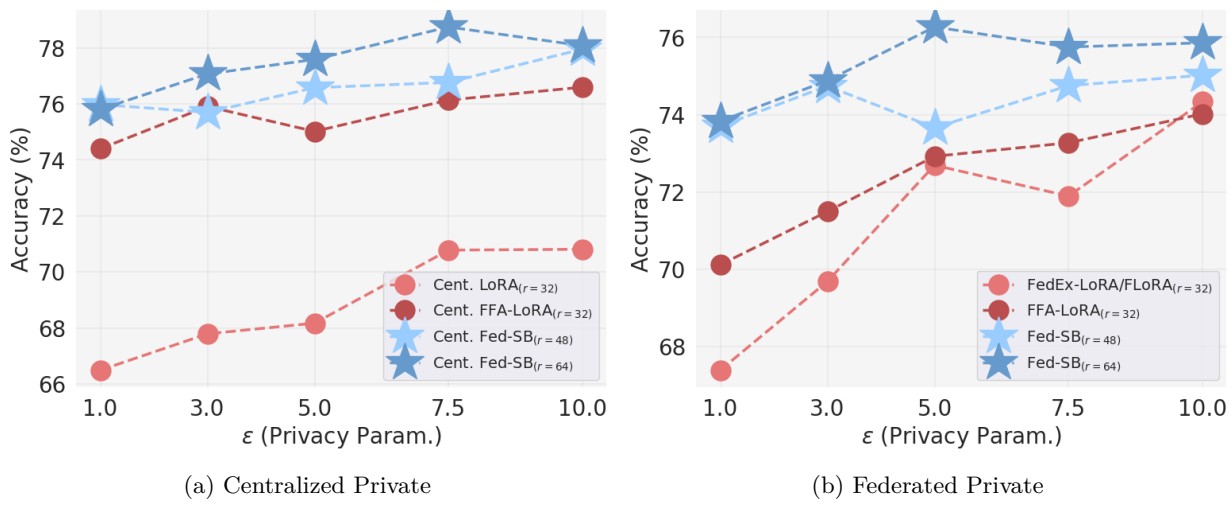

Figure 4: Performance vs. number of communicated parameters (in log scale) for various methods in federated fine-tuning across multiple models on arithmetic and commonsense reasoning tasks.

Figure 5: Performance comparison of various methods in centralized (Cent.) private and federated private fine-tuning (BERT-base) on SNLI across varying values of $\epsilon$.

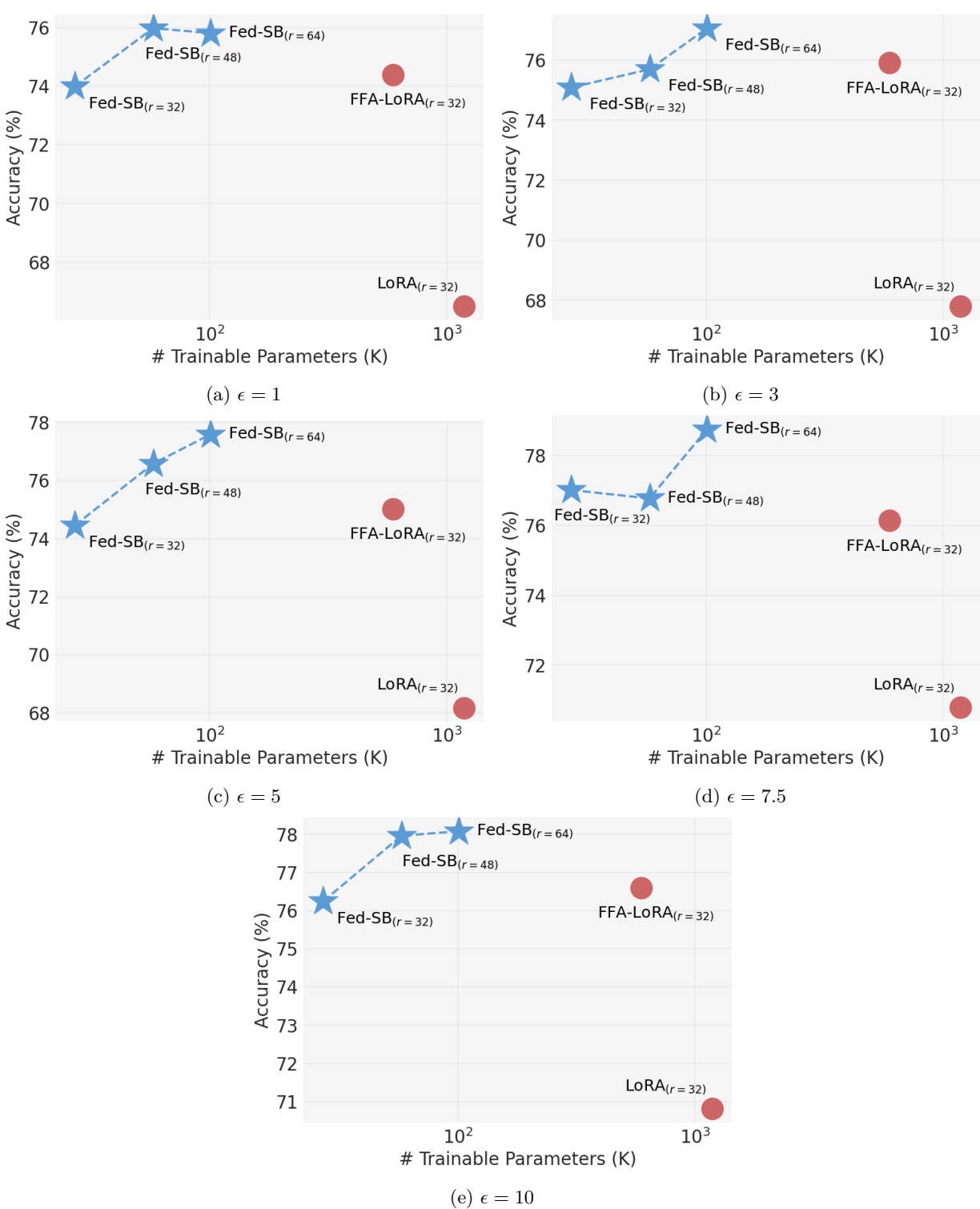

Figure 6: Performance vs. number of trainable parameters (in log scale) for various methods in centralized private fine-tuning (BERT-base) across different privacy budgets ($\epsilon$).

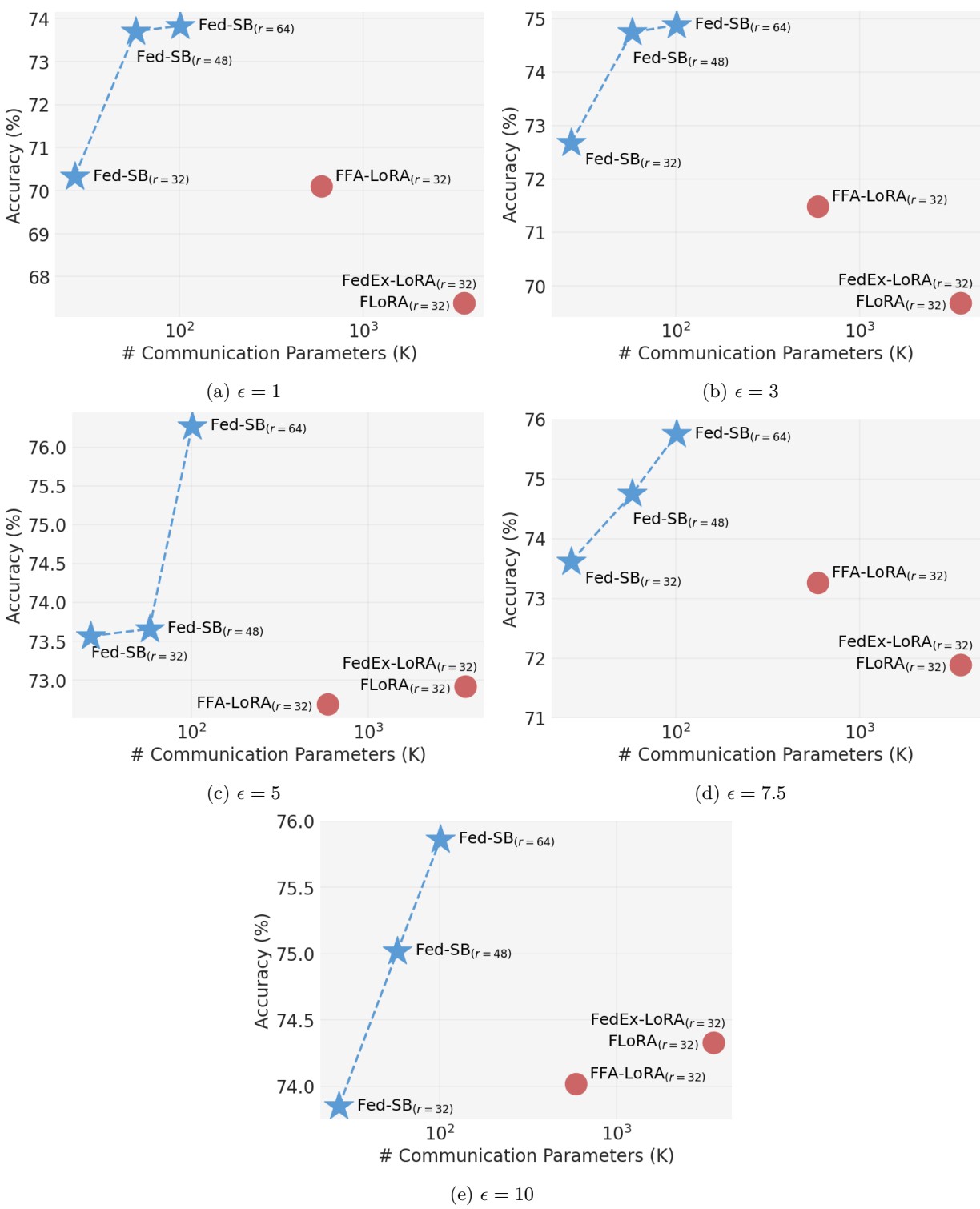

Figure 7: Performance vs. number of communicated parameters (in log scale) for various methods in federated private fine-tuning (BERT-base) across different privacy budgets ($\epsilon$).

