# OpenReview forum: "Fed-SB: A Silver Bullet for Extreme Communication Efficiency and Performance in (Private) Federated LoRA Fine-Tuning"
_TMLR — Accepted by TMLR_

### Review · Reviewer_ktdP · 2025-12-17

**Summary Of Contributions:**

The paper proposes a reparametrisation of LoRA that is more suitable for federated learning. The paper claims that the work outperforms existing federated variants of LoRA while incurring lower overhead.

**Additional Comments:**

The paper is difficult to follow because of the sensationalised writing and visual clutter. "Silver bullets", "pushing pareto frontier", excessive bolding in every single paragraph. It reads like a press release.

**Audience:**

Yes

**Audience Explanation:**

Federated learning is an active area of machine learning research. Federated learning of performance efficient adapters for large models naturally follows, given their popularity in centralised customisation of (large) language models.

**Claims And Evidence:**

No

**Claims Explanation:**

1. The empirical evidence is not very robust as it stands. The paper states that the results are an average of three runs. It is not enough runs to get a stable result. Additionally, the results should be reported as mean+std and not just the average. Furthermore, for the most all results are quite close; hence, depending on the standard deviations they might be statistically insignificant, and that should be tested for and reported.
2. The method outperforms everything else even at significantly lower capacity. This casts doubt whether all methods have been properly trained and not over/under-trained. The paper could really use some presentation of the loss landscape and convergence analysis.
3. Similarly, I'm not sure why such extensive discussion about the rank is needed. As the authors note, in standard LoRA the trainable part **W=AB** where the inner dimension *r* give us the efficiency, and here, it's replaced with **W=ARB** where **R** is *r x r*. Why do we care about it and not just the number of trainable parameters?
4. Given the abovementioned reduced capacity, I am not certain whether the selection of benchmarks faithfully represents the efficacy of the method. There should exist settings/benchmarks that require more trainable parameters than is afforded by a set of **R** matrices, which in turn would show how one needs to scale up **R** to meet them. This likely ties in with (1) and (2).
5. I don't understand why differential privacy (DP) is such a central part of the presentation of this work. It is imperative that mechanisms with fewer trainable parameters work better with DP. If we can trust all the results in this work (see above points), then it follows naturally that this method would be more suitable for private learning. However, I fail to see what algorithmic **specifically to private learning** this work introduces.

Without this information, I cannot make a definitive statement about the strengths of the paper where the core contribution boils down to training a smaller intermediate matrix in LoRA reparameterisation.

**Requested Changes:**

1. Loss landscape and convergence analysis for the compared methods
2. Example of a benchmark where a lower rank matrix cannot effectively learn the finetuning dataset due to insufficient capacity -- with experiments.
3. More runs with reported mean+std, statistical significance.
4. This paper has no algorithmic contribution to differential privacy. It boils down to "lower rank -> less noise -> better DP trade-off". The boilerplate can be moved to the appendix or just made into a 1-2 paragraph discussion piece, to make space for (1).
5. Streamlined writing (see additional comments).

---

> ### Author Response · Authors · 2026-01-18
> **Rebuttal to Reviewer ktdP (Part 1)**
>
> Thank you for the detailed review. Below, we address each point raised in detail.
>
> `Streamlined writing`
>
> Thank you for these suggestions. In response to your comments, we have made the writing more streamlined and easier to follow. Specifically, in the updated manuscript:
>
> - We removed mentions of improving the Pareto frontier.
> - We reduced excessive mentions of "silver bullet".
> - We removed all excessive bolding throughout the manuscript.
>
> We hope this makes our paper easier to read and follow.
>
>
>
> `W1: “The empirical evidence is not very robust as it stands. The paper states that the results are an average of three runs… Additionally, the results should be reported as mean+std and not just the average.”`
>
> Thank you for this suggestion. Following your comment, we report standard deviation values for all our experiments in **Appendix I**. These results show that our findings are robust and statistically significant.
> We would like to note that running multiple experiments at LLM scale is very expensive computationally and time-wise. Following most prior work in this area, results are reported as an average of 1-3 runs. We also do not have access to a large number of GPUs, all our results were run on a single A6000 GPU. We believe that averaging results over 3 runs is thus a very fair approach that aligns well with the overall efficient fine-tuning literature.
>
>
>
> `W3: “Similarly, I'm not sure why such extensive discussion about the rank is needed. As the authors note, in standard LoRA the trainable part W=AB …”`
>
> Rank is a fundamental capacity knob in low-rank adaptation that is not fully captured by parameter count. In standard LoRA, the update is parameterized as
> $$
> \Delta W = BA, \quad A \in \mathbb{R}^{r \times n}, \; B \in \mathbb{R}^{m \times r},
> $$
> which implies $\mathrm{rank}(\Delta W) \le r$. Thus, $r$ directly upper-bounds the intrinsic dimension of the learned update and governs whether the adapter underfits. Multiple recent PEFT works explicitly identify rank deficiency as a primary reason for degraded LoRA performance and motivate mechanisms to increase effective rank / mitigate underfitting, e.g., HiRA [1], MoRA [2], and LoRA-SB [3]. These works show that two methods with similar parameter budgets can behave very differently depending on how they address the rank/underfitting bottleneck; hence, parameter count alone is not a sufficient descriptor of adaptation expressivity.
>
> In our setting, Fed-SB uses the LoRA-SB parameterization
> $$
> \Delta W = B R A, \quad R \in \mathbb{R}^{r \times r},
> $$
>
> which still satisfies $\mathrm{rank}(\Delta W) \le r$. Therefore, discussing rank remains meaningful: it controls the update rank (capacity) in the same way as in LoRA, and it is the standard hyperparameter used across the low-rank adaptation literature and baselines.
>
> Crucially, Fed-SB also differs from "train $r^2$ parameters" in a way that makes naive parameter-count comparisons potentially misleading: the factors $A$ and $B$ are not randomly initialized but are constructed via a structured initialization / preconditioning (as described in LoRA-SB). This choice defines a well-conditioned adaptation subspace in which learning $R$ avoids the typical underfitting behavior associated with small-rank LoRA. As a result, the *effective* adaptation capacity is governed jointly by (i) the rank $r$ and (ii) the choice of the fixed basis $(A,B)$, not solely by the number of trainable entries in $R$.
>
> Finally, in federated learning the most relevant resource is communication, and rank is the natural knob that governs it for our method: Fed-SB communicates only $R$ (size $r^2$) per adapted weight, while $A$ and $B$ are fixed/shared and therefore do not contribute to per-round communication. This is precisely why we discuss rank extensively: it simultaneously controls adaptation capacity and the communicated payload, and the key architectural advantage of Fed-SB is that it enables competitive performance while keeping communication extremely low.

---

> > ### Author Response · Authors · 2026-01-18
> > **Rebuttal to Reviewer ktdP (Part 2)**
> >
> > `W4: “Given the abovementioned reduced capacity, I am not certain whether the selection of benchmarks faithfully represents the efficacy of the method…”`
> >
> > We agree with the underlying principle (capacity can be a bottleneck), but we believe the current benchmarks already instantiate exactly this phenomenon in the PEFT literature. For instance, on commonsense reasoning, standard LoRA is widely observed to be rank-limited, and increasing the LoRA rank yields substantial accuracy gains.
> >
> > For example, HiRA [1] explicitly motivates high-rank adaptation using a commonsense-reasoning setup and shows that moving from commonly used low ranks (e.g., $r=32$) to much higher ranks (e.g., $r=512$ and beyond) markedly improves accuracy across the constituent datasets (see Figure 2 and the accompanying discussion).
> >
> > In the same section, HiRA also notes that despite these gains, practical PEFT often defaults to low ranks such as 32 or 64 due to efficiency constraints, even though higher ranks can be beneficial on complex tasks.
> >
> > This directly addresses the concern that the benchmarks are too easy: commonsense reasoning is a canonical example where low-rank adapters underfit and scaling rank improves performance. Our setting focuses on achieving competitive performance under stringent federated communication constraints, i.e., enabling the use of sufficiently expressive adaptations (e.g., higher effective rank where needed) while keeping the communicated payload extremely small. In that sense, these benchmarks are representative and also aligned with the motivating failure mode in PEFT. The tasks are the ones where one would want to increase adaptation capacity but are prevented from doing so by efficiency/communication limits.
> >
> >
> > `W2: “The method outperforms everything else even at significantly lower capacity. This casts doubt whether all methods have been properly trained and not over/under-trained…”`
> >
> > Our experimental setup follows the standard training protocols used in the federated PEFT literature, and in particular we adopt the same optimization choices (optimizer, schedules, local steps/epochs, etc.) as the prior methods we compare against (e.g., FLoRA and FedEx-LoRA). Thus, the observed gains are not due to an unusual training design that disadvantages baselines.
> >
> > To directly address the convergence concern, we provide training-loss-versus-steps curves (for a representative client/representative run) in **Appendix F (Figure 3)** showing that all methods exhibit stable optimization and converge under the same training budget, i.e., there is no evidence that the baselines are under-trained or over-trained. Fed-SB typically reaches a lower training loss earlier and/or achieves a better final loss, consistent with improved optimization dynamics rather than a mismatch in training effort.
> >
> > Finally, there are principled architectural reasons why Fed-SB can converge more favorably in federated settings. In LoRA-style federated training, aggregating factorized updates can introduce an approximation or bias because averaging factors is not equivalent to averaging the resulting weight updates. Fed-SB avoids this issue because the communicated variable is aggregated exactly in the adapter space (linear averaging in $R$), and all clients share a common, preconditioned basis via fixed projectors $A$ and $B$. These properties reduce aggregation-induced error and yield more consistent global updates, which provide a natural explanation for the improved convergence behavior observed empirically.

---

> ### Author Response · Authors · 2026-01-18
> **Rebuttal to Reviewer ktdP (Part 3)**
>
> `W5: “I don't understand why differential privacy (DP) is such a central part of the presentation of this work….”`
>
>
> A core premise of FL is that clients do not share raw data due to privacy concerns, and it is well-established that trained models can still leak information about their training data. Therefore, evaluating and discussing privacy in FL settings is natural and commonly expected.
>
> Prior work in federated PEFT has explicitly treated privacy as a first-class goal and proposed architectural modifications to improve privacy. A prominent example is FFA-LoRA [4], which we compare against. FFA-LoRA is motivated in part by privacy and reduces leakage risk by restricting what is trained and communicated. In the same spirit, our work provides an architectural route to improved privacy properties in federated low-rank adaptation; however, we emphasize that this is not the only contribution, our primary thrust remains competitive performance under extremely low communication.
>
> Further, our privacy-related argument is not merely that fewer parameters help DP. Beyond parameter reduction, Fed-SB changes the structure of the trainable object in a way that matters for private optimization. Standard LoRA learns two factors and is bilinear in the trainable parameters, so under DP-SGD noise injection one can obtain a cross-product noise term (i.e., a second-order interaction between the noises on the two factors). In contrast, Fed-SB trains only a single matrix $R$ in
> $\Delta W = B R A,$ with $A,B$ fixed and shared, so the private perturbation enters linearly through $R$ and avoids the bilinear cross-product noise interaction present when both factors are trained. This is an algorithmic and architectural distinction that affects privacy-utility behavior beyond the parameter count.
>
> Finally, our approach is complementary and orthogonal to existing DP improvements. Fed-SB can be combined with standard DP techniques (e.g., improved clipping strategies, privacy accounting, optimizers, or secure aggregation mechanisms) because it primarily changes the adapter parameterization and the communicated object, rather than relying on a specialized privacy mechanism that would preclude other advances.
>
>
> ---
>
> ### References
>
> [1] HiRA: Parameter-Efficient Hadamard High-Rank Adaptation for Large Language Models, https://openreview.net/forum?id=TwJrTz9cRS
>
> [2] MoRA: High-Rank Updating for Parameter-Efficient Fine-Tuning, https://arxiv.org/abs/2405.12130
>
> [3] Initialization using Update Approximation is a Silver Bullet for Extremely Efficient Low-Rank Fine-Tuning, https://arxiv.org/abs/2411.19557
>
> [4] Improving LoRA in Privacy-preserving Federated Learning, https://arxiv.org/abs/2403.12313
>
> ---
>
> Thank you again for your detailed review. Please let us know if you have any further questions or need additional clarification.

---

### Review · Reviewer_PLri · 2025-12-28

**Summary Of Contributions:**

The paper provides a method for achieving exact gradient updates in parameter efficient fine-tuning for LLMs. The key idea is to build upon a recent work LoRA-SB whose structure is compatible with exact gradient updates compared to prior works. While LoRA-SB considers a centralized setting the application to the federated setting is natural and appears to provide significant gains in communication efficiency over prior approaches. The authors demonstrate that the proposed method is also a natural candidate for introducing differential privacy during the training process.

**Additional Comments:**

The proposed work seems to be a natural extension of prior work on LoRA-SB to the federated setting. The the empirical results look interesting. I would request the authors to address the questions raised above.

**Audience:**

Yes

**Audience Explanation:**

The topics on federated fine-tuning of LoRA models and differential privacy are important and timely and of interest to the TMLR audience.

**Broader Impact Concerns:**

none that I can consider.

**Claims And Evidence:**

Yes

**Claims Explanation:**

While the extension of LoRA-SB to the federated setting is natural, the paper provides a comprehensive set of experiments showing impressive performance gains. In particular the non-iid setting is handled in a natural way and shows impressive gains.

**Requested Changes:**

Lemma 1 is the only theoretical proof in the paper and the proof is unfortunately not very clear.
- how do you go from (20) to (21)? What is the choice of alpha used?
- in practice if the learning parameters are reduced why should other parameters e.g. T, q etc be the same accross different schemes?
I am not an expert in the topic of differential privacy and I do not understand the whole proof. It is not written in a very rigorous manner. Some variables do not seem to be defined. What is \sigma_noise?

Why is the initialization used similar to what is done in LoRA-SB?  If you have different heterogenous clients, better intializations could be used to improve the performance.

There are no error bars presented in the experimental results. Are the results statistically significant?

Can you provide any theoretical convergence guarantees especially in the non-iid setting to demonstrate further novelty over prior works?

---

> ### Author Response · Authors · 2026-01-18
> **Rebuttal to Reviewer PLri (Part 1)**
>
> Thank you for your positive assessment and detailed review. Below, we answer each of the points raised by you.
>
> `Improving clarity around Lemma 1`
>
> We thank the reviewer for pointing this out. Below we clarify (i) how Eq. (20) implies Eq. (21), including the choice of the RDP order $\alpha$, (ii) what $\sigma_{\text{noise}}$ denotes, and (iii) why $T,q$ are held fixed in the statement.
>
> ---
>
> ### (i) How do we go from (20) to (21)? What is the choice of $\alpha$?
> Eq. (20) has the standard form obtained when converting a composed RDP guarantee to $(\varepsilon,\delta)$-DP:
> $$
> \varepsilon(\alpha) = O\left( A\alpha + \frac{B}{\alpha-1} \right), \quad \text{where } A := \frac{q^2 T}{\sigma^2}, B := \log(1/\delta).
> $$
> To obtain the tightest (smallest) $\varepsilon$, one optimizes over $\alpha>1$ (this is standard in RDP-to-DP conversions). Minimizing
> $$
> f(\alpha) = A\alpha + \frac{B}{\alpha-1}
> $$
> gives the choice
> $$
> \alpha^\star - 1 = \Theta\left(\sqrt{\frac{B}{A}}\right)
> = \Theta\left(\sqrt{\frac{\log(1/\delta)\,\sigma^2}{q^2 T}}\right).
> $$
> Substituting $\alpha^\star$ back yields
> $$
> \varepsilon = O\left(\sqrt{AB}\right)
> = O\left(\frac{q\sqrt{T\log(1/\delta)}}{\sigma}\right).
> $$
> Finally, using the scaling assumption in the appendix text that $\sigma \propto 1/\sqrt{d}$, Eq. (*) becomes
> $$
> \varepsilon = O\left(q\sqrt{Td\\log(1/\delta)}\right)
> = O(\sqrt{d}),
> \tag{21}
> $$
> which matches Eq. (21). (Up to universal constants hidden in the big-$O$.)
>
> Thus, the step (20)$\to$(21) is obtained by the *standard optimization over $\alpha$* in the RDP conversion, yielding $\varepsilon = O\left(q\sqrt{T\log(1/\delta)}/\sigma\right)$, and then substituting $\sigma \propto 1/\sqrt{d}$.
>
> ---
>
> ### (ii) What is $\sigma_{\text{noise}}$?
> In DP-SGD, after per-example gradient clipping to norm $C$, one adds isotropic Gaussian noise:
> $$
> \mathcal{N}\left(0; \sigma^2 C^2 I_d\right).
> $$
> Here $\sigma$ is the **noise multiplier** (dimensionless), and the **actual standard deviation** of the added noise per coordinate is
> $$
> \sigma_{\text{noise}} = \sigma C.
> $$
> This is why the appendix states $\sigma_{\text{noise}}=\sigma C$ when plugging into the Gaussian mechanism RDP bound
> $$
> \varepsilon_{\text{RDP}}(\alpha) = \frac{\alpha\,\Delta_2^2}{2\sigma_{\text{noise}}^2},
> $$
> with $\Delta_2=C$ for clipped gradients.
>
> ---
>
> ### (iii) Why should $T,q$ (and other quantities) be the same across schemes?
> Lemma 1 is a **scaling statement under a fixed training protocol**: for DP-SGD with given sampling probability $q$, number of steps $T$, clipping norm $C$, and noise multiplier $\sigma$, the privacy loss obeys the stated dependence on the number of learnable parameters $d$ (via the substitution $\sigma \propto 1/\sqrt{d}$ used in the sketch).
>
> In practice, different schemes could indeed choose different $(T,q,C,\sigma)$ depending on optimization needs. The lemma does **not** claim that $T$ and $q$ must always be identical across all algorithms; rather, it isolates the effect of changing the learnable dimension $d$ **holding the rest of the DP-SGD setup fixed**, which is the clean way to reason about how parameter-efficiency impacts the privacy-utility tradeoff.
>
> Equivalently, one can read the same scaling in the reverse direction: for a *fixed* target $(\varepsilon,\delta)$, $q$, and $T$, the required noise level scales as
> $$
> \sigma=\Theta\left(\frac{q\sqrt{T\log(1/\delta)}}{\varepsilon}\right),
> $$
> and if the effective $\sigma$ required by the learning dynamics scales like $1/\sqrt{d}$, then $\varepsilon$ scales like $O(\sqrt{d})$.
>
> `W2: “Why is the initialization used similar to what is done in LoRA-SB? If you have different heterogenous clients, better initializations could be used to improve the performance.”`
>
> We appreciate this thoughtful question.
>
> The choice of LoRA-SB initialization is driven by our exact aggregation requirement.
> If we initialized $A$ and $B$ matrices differently across clients using local datasets in the data-heterogenous setting, each client would have distinct $A_i$ and $B_i$ matrices. This would break exact aggregation via simple $R$ matrix averaging.
>
> To aggregate heterogeneous low-rank subspaces exactly, we would need to either: (1) transmit full $A$ and $B$ matrices from all clients, or (2) employ expensive subspace alignment procedures, both incurring significantly higher communication/computational costs that undermine the efficiency benefits of our method.
>
> The uniform LoRA-SB initialization maintains exact aggregation while achieving strong performance across heterogeneous settings (Table 3).
> While client-specific initializations could offer improvements, they would require sacrificing exact aggregation or substantially increasing costs. We leave investigating such trade-offs as valuable future work, particularly for scenarios where approximate aggregation may be acceptable.

---

> > ### Author Response · Authors · 2026-01-18
> > **Rebuttal to Reviewer PLri (Part 2)**
> >
> > `W3: “There are no error bars presented in the experimental results. Are the results statistically significant?”`
> >
> > Thank you for bringing this up. Following your comment, we now report standard deviation values for all our experiments in **Appendix I**. Our results are reported as an average of 3 runs along with their standard deviations. These results demonstrate that our findings are robust and statistically significant.
> >
> >
> > `W4: “Can you provide any theoretical convergence guarantees especially in the non-iid setting to demonstrate further novelty over prior works?”`
> >
> >
> > We do not claim any new theoretical convergence guarantees beyond what is known in the federated optimization literature, and our contribution is primarily architectural/algorithmic (achieving a strong accuracy–communication tradeoff).
> >
> > That said, our design explicitly targets properties that prior theory and practice associate with improved convergence behavior in federated learning, especially under non-IID data.
> >
> > 1. **Exact aggregation and convergence.** A key distinction of Fed-SB is that the server aggregates the *actual* client updates exactly (in the adapter space), rather than aggregating a factorization whose product differs from the average update. Prior work has argued that such *approximate aggregation* can introduce a persistent bias/error term in the global update and thereby harm convergence. For example, recent analyses [1] motivate the importance of exact (or unbiased) aggregation for stable convergence in federated settings. Fed-SB satisfies this property by construction through the shared fixed basis and linear averaging in the communicated variable.
> >
> > 2. **DP noise structure and optimization stability.** In differentially private training, optimization is impacted not only by the magnitude of noise but also by whether the parameterization induces higher-order noise interactions. In bilinear parameterizations (e.g., training both LoRA factors), DP perturbations can yield cross-product noise terms, whereas in Fed-SB the injected noise enters linearly through the trained matrix $R$. This reduces effective noise interactions and is a strong heuristic for improved optimization stability, which is particularly relevant in non-IID regimes where optimization is already harder due to client drift.
> >
> > In summary, while we do not provide a new formal convergence theorem, Fed-SB is constructed to satisfy properties that are linked to convergence in existing theory and that practitioners rely on when reasoning about non-IID federated optimization: (i) exact/unbiased aggregation of the learned update and (ii) a parameterization that avoids bilinear noise amplification under DP. We view these as principled architectural choices grounded in prior analyses rather than new theoretical claims.
> >
> > ---
> >
> > ### References
> > [1] Exploring Gradient Subspaces: Addressing and Overcoming LoRA's Limitations in Federated Fine-Tuning of Large Language Models, https://arxiv.org/abs/2410.23111
> >
> > ---
> >
> > Thank you again for your detailed review. We would be happy to clarify any remaining questions you might have.

---

### Review · Reviewer_3AEH · 2026-01-17

**Summary Of Contributions:**

Exact aggregation with minimal communication: By training only a small r×r matrix R while freezing adapters  A and B, Fed-SB enables exact model aggregation via simple averaging, reducing communication costs by up to 230× compared to prior methods.

Enhanced privacy preservation: The reduced parameter count lowers the noise required for differential privacy, and the linear update structure avoids noise amplification—a common issue in bilinear LoRA-based methods.

Strong empirical performance: Extensive evaluations on commonsense reasoning, arithmetic reasoning, and language inference tasks show that Fed-SB outperforms existing federated fine-tuning methods while being significantly more communication-efficient.

Extensions to heterogeneous settings: The method is adapted to support clients with different rank budgets, maintaining exact aggregation and robust performance.

**Audience:**

Yes

**Audience Explanation:**

1.Clear problem framing: The paper nicely motivates the trade-offs between exact aggregation, communication cost, and expressivity in federated LoRA.

2.Privacy-aware design: The integration of differential privacy is thoughtful, with analysis showing how Fed-SB naturally reduces noise injection and avoids second-order noise terms.

3.Practical insights: The inclusion of memory and training time analyses strengthens the practical relevance of the work.

**Broader Impact Concerns:**

I have not found any ethical concerns.

**Claims And Evidence:**

Yes

**Claims Explanation:**

The study covers multiple model families (BERT, Mistral, Gemma, Llama), diverse benchmarks, and both IID and non-IID data distributions.

**Requested Changes:**

The authors may consider discussing the following related works to provide additional context and strengthen the privacy and robustness perspectives:

For privacy-aware mean estimation and user-level DP:

[1] Zhao, Puning, Lifeng Lai, Li Shen, et al. "A huber loss minimization approach to mean estimation under user-level differential privacy." Advances in Neural Information Processing Systems 37 (2024): 130018-130056.*
This work offers relevant insights into robust and private aggregation, which could complement the differential privacy analysis in Fed-SB.

For Byzantine-resilient federated learning under data heterogeneity:

[2] Zuo, Shiyuan, Xingrun Yan, Rongfei Fan et al. "Federated learning resilient to byzantine attacks and data heterogeneity." IEEE Transactions on Mobile Computing (2025).

A brief discussion of  this paper would help position Fed-SB within the broader literature on robust federated learning, especially regarding adversarial client scenarios.

---

> ### Author Response · Authors · 2026-01-18
> **Rebuttal to Reviewer 3AEH**
>
> Thank you for your positive assessment and detailed analysis of our paper. We appreciate your constructive feedback. Here is our response to your suggestion.
>
> `The authors may consider discussing the following related works to provide additional context and strengthen the privacy and robustness perspectives...`
>
> Thank you for pointing out these relevant papers. We have incorporated both works into the Related Work section of the updated manuscript to strengthen the robustness and privacy analysis as you suggested.
>
> ---
>
> We appreciate your thoughtful review and are happy to provide any additional clarifications you may need.

---

### Decision · Action_Editor_zvNb · 2026-03-02

**Recommendation:** Accept as is

**Audience:**

Yes

**Audience Explanation:**

The work is about efficient federated fine-tuning of LLMs, which is relevant for the TMLR audience.

**Claims And Evidence:**

Yes

**Claims Explanation:**

The authors empirically demonstrate good performance and communication-efficiency on a variety of models and tasks.